# Probiotic consumption influences universal adaptive mutations in indigenous human and mouse gut microbiota

Chenchen Ma[1,5], Chengcheng Zhang[2,5], Denghui Chen[3,5], Shuaiming Jiang[1], Siyuan Shen[1], Dongxue Huo[1], Shi Huang 🔟 [4✉], Qixiao Zhai🔟 [2✉] & Jiachao Zhang🔟 [1✉]

The adaptive evolution in indigenous intestinal microbes derived from probiotics is critical to safety and efficacy evaluation of probiotics, yet it is still largely underexplored. Here, through 11 publicly accessible datasets, we demonstrated that probiotic consumption can lead to widespread single-nucleotide variants (SNVs) in the native microbiota. Interestingly, the same probiotic strains introduced far more SNVs in mouse gut than humans. Furthermore, the pattern of probiotics-induced SNVs was highly probiotic-strain specific, and 17 common SNVs in *Faecalibacterium* prausnitzii genome were identified cross studies, which might lead to changes in bacterial protein structure. Further, nearly 50% of *F. prausnitzii* SNVs can be inherited for six months in an independent human cohort, whereas the other half only transiently occurred. Collectively, our study substantially extended our understanding of co-evolution of the probiotics and the indigenous gut microbiota, highlighting the importance of assessment of probiotics efficacy and safety in an integrated manner.

[1] College of Food Science and Engineering, Key Laboratory of Food Nutrition and Functional Food of Hainan Province, Hainan University, 570228 Haikou, China. [2] State Key Laboratory of Food Science and Technology, School of Food Science and Technology, Jiangnan University, 214122 Wuxi, China. [3] Department of Psychiatry, University of California San Diego, La Jolla, CA 92093, USA. [4] Department of Pediatrics and Center for Microbiome Innovation at Jacobs School of Engineering, University of California San Diego, 9500 Gilman Drive, La Jolla, CA 92093, USA. [5] These authors contributed equally: Chenchen Ma, Chengcheng Zhang, Denghui Chen. ✉email: shihuang047@gmail.com; zhaiqixiao@sina.com; zhjch321123@163.com

Gut microbiota has been widely implicated in many host diseases and shown to provide potential microbial targets for therapeutic development, such as diabetes[1], inflammatory bowel disease[2], and irritable bowel syndrome[3]. Probiotics are live microorganisms that can enhance host health by modulating the gut microbiome while administered in adequate amounts (FAO/WHO)[4,5]. Dietary administration of probiotics[6] has been accepted as one of the most important strategies to modulate the gut microbiota for human health.

The ecological effect of probiotic administration on gut microbiota composition has been well documented in previous clinical microbiome studies or animal models[7–10]. However, evolutionary pressures leading to changes, such as single-nucleotide variants (SNVs), in the indigenous gut microbial community, thereby altering the functional potential of the gut microbiome. Many studies implied that a small number of genetic mutations or even a single SNV in the microbial genome can significantly alter the pathogenic behavior of gut bacteria and affect host health[11–13]. Chen et al.[1] identified that specific SNVs on the genome of *Bacteroides coprocola* were correlated with T2D. Zou et al.[12] reported that BlcE84-encoding bacteria with a distinctive SNV on the genome caused the destruction of the worm and mouse epithelial barrier and immune activation. Bacterial genetic variations in specific locations can even promote the longevity of their host[14].

A previous study has highlighted that the evolution and transfer of genetic information of host-associated microbiota could enable resilience to biotic and abiotic perturbations[15]. In the organism *Bacteroides fragilis*, for example, many parallel evolutions of genes were found related to cell-envelope biosynthesis and polysaccharide utilization[16]. Notably, probiotics can impose persistent selective pressures on host gut bacteria by accumulating mutations related to carbohydrate utilization and acid tolerance within the mouse gut microbiome, such as *E. coli*. Nissle[17] and the driving force may provide the potential for genomic variations of gut resident species[16]. The indigenous gut microbiota, including competitors and collaborators, rapidly evolved to adapt to the ecological invasion of probiotics[18]. However, these in vivo genetic processes of gut microbiota are still poorly characterized due to probiotic consumption using a wide array of human and animal models. The in vivo evolution of the indigenous gut microbiota and probiotics facilitates the understanding to leverage these gut selective forces for the genetic engineering of probiotics. Hence, the comprehensive analysis of genomic alterations in gut commensal bacteria after probiotic exposure was important for evaluating the safety of probiotics and investigating the long-term effect of probiotics on the functional dynamics of host gut commensal bacteria. Adaptive evolution of gut microbes can be confirmed by parallel evolution or convergent evolution or by increased frequency of mutations inconsistent with neutral drift[16]. Shotgun metagenomic sequencing technologies provide access to the entire gut microbiome genetic information and thus enable such a thorough investigation of adaptive SNVs arisen by probiotic ingestion and microbial composition and functional genes involved.

This meta-analysis aims to systematically evaluate the effect of probiotic administration on strain-level variations of the gut microbiome, and the association between adaptive SNVs, host species, probiotic strains, probiotic intervention duration, and probiotic dose. Furthermore, we also sought to identify and characterize the universal adaptive mutations arisen from probiotic ingestion in a wide range of shotgun metagenomic studies.

## Results

### Probiotic intake commonly altered the genetic composition of gut microbial residents. To comprehensively understand the

adaptive mutations in the resident gut microbiota due to probiotic intake, we first collected and curated publicly available metagenomic studies related to probiotics with the following criteria. (1) The study has a longitudinal design, which at least has a baseline and end time point for the probiotic consumption for a human or animal host subject. (2) The study does not use probiotics in combination with any other substance, such as medications, prebiotics, minerals, vitamins. (3) The study's raw data were published and had detailed metadata. (4) The study's sequencing data quality allows us to analyze at least species-level composition in the gut microbiome. (5) The study provided clear probiotics species/strains/product information. (6) The study has a clear statement on the dose and duration for probiotic intake. Finally, 11 high-quality metagenomic studies were included, among which seven were human cohorts (U.S., $N = 1$; Israel, $N = 1$; New Zealand, $N = 2$; China, $N = 3$), and five were animal cohorts (dog, $N = 1$; rat, $N = 1$; mice, $N = 3$). In total, 224 probiotic-treated individuals and 197 placebo controls (Tables 1 and 2) were included. The probiotic-administration duration for hosts in studies ranged from 1 week to 2 years and its median was 4 weeks. The median dose of probiotic administration was $9^{10}$ CFU/day, ranging from $10^8$ to $10^{10}$. Next, MetaPhlan2 was employed to identify the microbial compositions that have a relative abundance >0.5% for SNV profiling (Supplementary Data 1). The metagenomic reads were then mapped to the reference genomes of these selected species for SNV identification. We compare the SNVs against each reference genome for each host before and after probiotic treatment. A total of 16,901 SNVs were associated with probiotic administration (Supplementary Data 2). We first wondered how diverse were resident gut microbes that spontaneously mutated after probiotic consumption and if such diversity can be different from usual (the control group). Interestingly, the number of gut resident species occurring SNVs significantly decreased with hosts after the dietary intervention with Probio-Fit, *Lactobacillus rhamnosus GG* (*L. rhamnosus* GG) and *Bifidobacterium lactis* HN019, besides in the mice with *Lactobacillus plantarum* HNU082 (*L. plantarum* HNU082) (Wilcoxon rank-sum test, Fig. 1a). Next, raw SNV frequency might be not comparable across studies due to the inevitable sample/study-level disparity in the metagenome sequencing depth. Specifically, raw SNV frequency positively correlated with sequencing depth in both human and mice populations (Fig. 1b and Supplementary Fig. 1). To reduce this technical bias across studies, a sequencing-depth-normalized number of SNVs (nSNVs) was used for the following cross-study comparisons.

$$nSNVs = the\ number\ of\ SNVs/sequencing\ depth\ per\ sample$$

(1)

The consumption of probiotic *L. plantarum* HNU082, *L. rhamnosus* GG, and *Bifidobacterium lactis* HN019 significantly reduced the total frequency of SNVs (nSNVs) in the gut residents (Wilcoxon rank-sum test, Fig. 1c). Overall, these suggested that probiotic intake can significantly change the genetic composition of a wide range of indigenous gut microbiota that was often not assumed.

### The nSNVs introduced by probiotics consumption were strain-specific. We next compared the nSNVs before and after probiotic intake in each of the studies. Overall, probiotic intake caused more SNVs in gut microbiota than the control group without any probiotic consumption (Fig. 2a). Furthermore, the SNVs in mice native gut microbiome outnumbered that in humans. Next, alpha diversities (Shannon and Simpson index) and beta diversity were

**Table 1 Animal fecal metagenomic studies of the probiotic intervention included in this meta-analysis.**

| Animal | Interventions | Dose (CFU/d) | Duration | Fecal sample size | DNA preparation | Sequencing platform | Average sequencing target depth (GB) | Read length (bp) | Sequencing data |
|---|---|---|---|---|---|---|---|---|---|
| C57BL6/N mice | Probiotic mixture | $10^9$ | 38 days | 4 | EZ1 Virus Mini kit v 2.0 (Qiagen) | Illumina HiSeq 2000 | 4.62 | 100 | SRP062583 |
| C57BL6/J mice | Lactobacillus rhamnosus GG | $10^8$ | 8 weeks | 2 | QIAamp Fast DNA Stool Mini Kit | Illumina HiSeq1500 | 5.74 | 100 | PRJNA293015 |
| Dog | Probio-Fit® | $2 \times 10^{10}$ | 60 days | 40 | Qiagen DNA Stool Mini-Kit | Illumina HiSeq XTEN | 2.84 | 150 | PRJNA524271 |
| Rat | Lactobacillus casei ATCC334 | $2 \times 10^8$ | 30 days | 14 | No mention | BGISEQ-500 | 7.19 | 100 | PRJEB22973 |
| C57BL6/J mice | Lactobacillus plantarum HNU082 | $4 \times 10^8$ | 4 weeks | 12 | QIAamp® DNA Stool Mini Kit (Qiagen) | Illumina HiSeq 2500 | 5.89 | 150 | PRJNA608079 |

**Table 2 Human fecal metagenomic studies of the probiotic intervention included in this meta-analysis.**

| Country | Interventions | Dose (CFU/d) | Duration | Fecal sample size | DNA preparation | Sequencing platform | Average sequencing target depth (GB) | Read length (bp) | Sequencing data |
|---|---|---|---|---|---|---|---|---|---|
| American | Bifidobacterium longum AH1206 | $10^{10}$ | 2 weeks | 42 | QIAamp® DNA Stool Mini Kit (Qiagen) | Illumina HiSeq 2500 | 2.74 | 100 | PRJNA324129 |
| Israel | Supherb Bio-25 | $2.5 \times 10^{10}$ | 4 weeks | 20 | DNeasy PowerLyzer PowerSoil Kit and PowerMag Soil DNA Isolation Kit | Illumina NextSeq | 1.19 | 80 | PRJEB28097 |
| New Zealand | Lactobacillus rhamnosus GG | $6 \times 10^9$ | 2 years | 129 | zirconium beads-beating | Illumina HiSeq 2500 | 3.42 | 125 | PRJNA345144 |
| New Zealand | Bifidobacterium lactis HN019 | $6 \times 10^9$ | 2 years | 24 | zirconium beads-beating | Illumina HiSeq 2500 | 2.85 | 125 | PRJNA345144 |
| China | Probio-Fit® | $5 \times 10^{10}$ | 30 days | 82 | BacteriaGen DNA kit (CW0552, CWBIO) | Illumina HiSeq 2500 | 7.51 | 150 | PRJNA554501 |
| China | Lactobacillus plantarum HNU082 | $7 \times 10^9$ | 7 days | 14 | QIAamp® DNA Stool Mini Kit (Qiagen) | Illumina HiSeq 2500 | 5.42 | 150 | PRJNA590026 |
| China | Lactobacillus casei Zhang | $10^{10}$ | 28 days | 40 | QIAamp® DNA Stool Mini Kit (Qiagen) | Illumina HiSeq 2500 | 7.05 | 150 | PRJNA762428 |

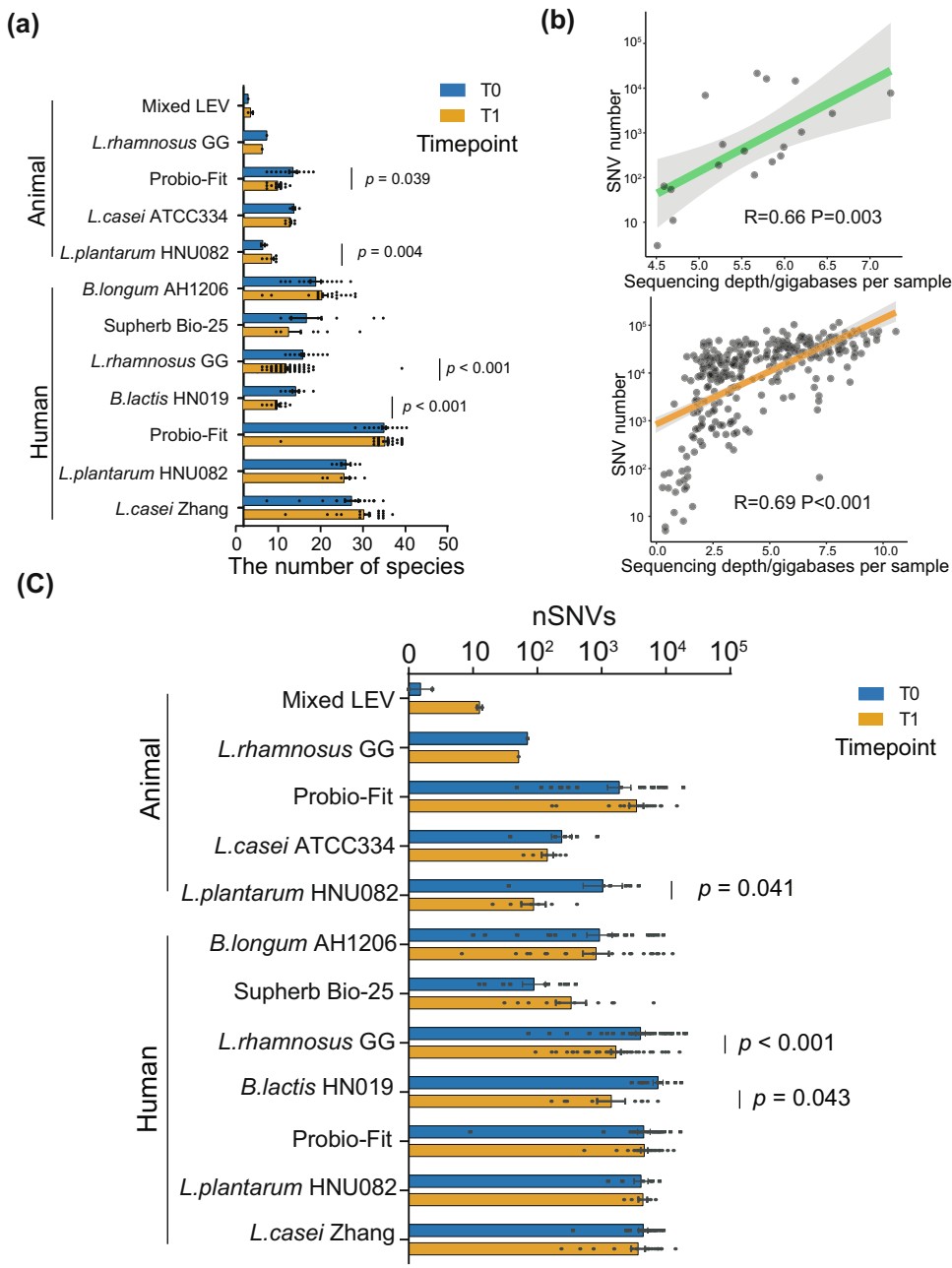

**Fig. 1 The alteration in the genetic diversity of native gut microbiome due to probiotic intervention. a** The barplot indicates the number of gut resident species that had SNVs after probiotic intake. The bars represent the number of species that had SNVs. **b** The scatter plot shows the correlation between the raw number of SNVs and the sequencing depth of the target genomes. Green represents the mice ($R = 0.66$, $P = 0.003$), while orange represents the human cohort ($R = 0.69$, $P < 0.001$). **c** The barplot indicates the normalized number of SNVs identified in two time points (baseline and end point of probiotic intervention in each study). All error bars represent the SEM. The source data for graphs are available as Supplementary Data 7 or FigShare (https://figshare.com/projects/Probiotic_consumption_influences_universal_adaptive_mutations_in_indigenous_human_and_mouse_gut_microbiota/122447).

calculated for each sample based on the profile of species-level SNVs. PERMANOVA was used to measure the effect size of probiotic intake on the SNV profiles at the species level ($p < 0.001$) (Fig. 2b−d and Supplementary Data 3). Our results suggested that the overall pattern of SNVs induced by probiotics was highly specific to probiotic strains. Furthermore, there is no significant correlation between nSNVs and experimental factors such as probiotics dose and duration of probiotics, observed from our investigation (Fig. 2e).

To mitigate the potential effect of confounding factors, such as individuality in the gut microbiome, in our analyses, six probiotic

studies were focused, including *Bifidobacterium longum* AH1206 (*B. longum* AH1206)[7], Supherb Bio-25 [19], *L. rhamnosus* GG[20], Probio-Fit[21], *L. plantarum* HNU082 [22] and *Lactobacillus casei* Zhang (*L. casei* Zhang) where host participants had paired/repeated microbiome measurements before and after the probiotic intervention. The correlation pattern between nSNVs and the beta diversity of gut microbiota was highly specific to what probiotic strains had been consumed (Fig. 2f). We found that nSNVs caused by *B. longum* AH1206 and *L. plantarum* HNU082 consumption had a positive correlation with Bray−Curtis distance of gut microbiota between baseline and post-

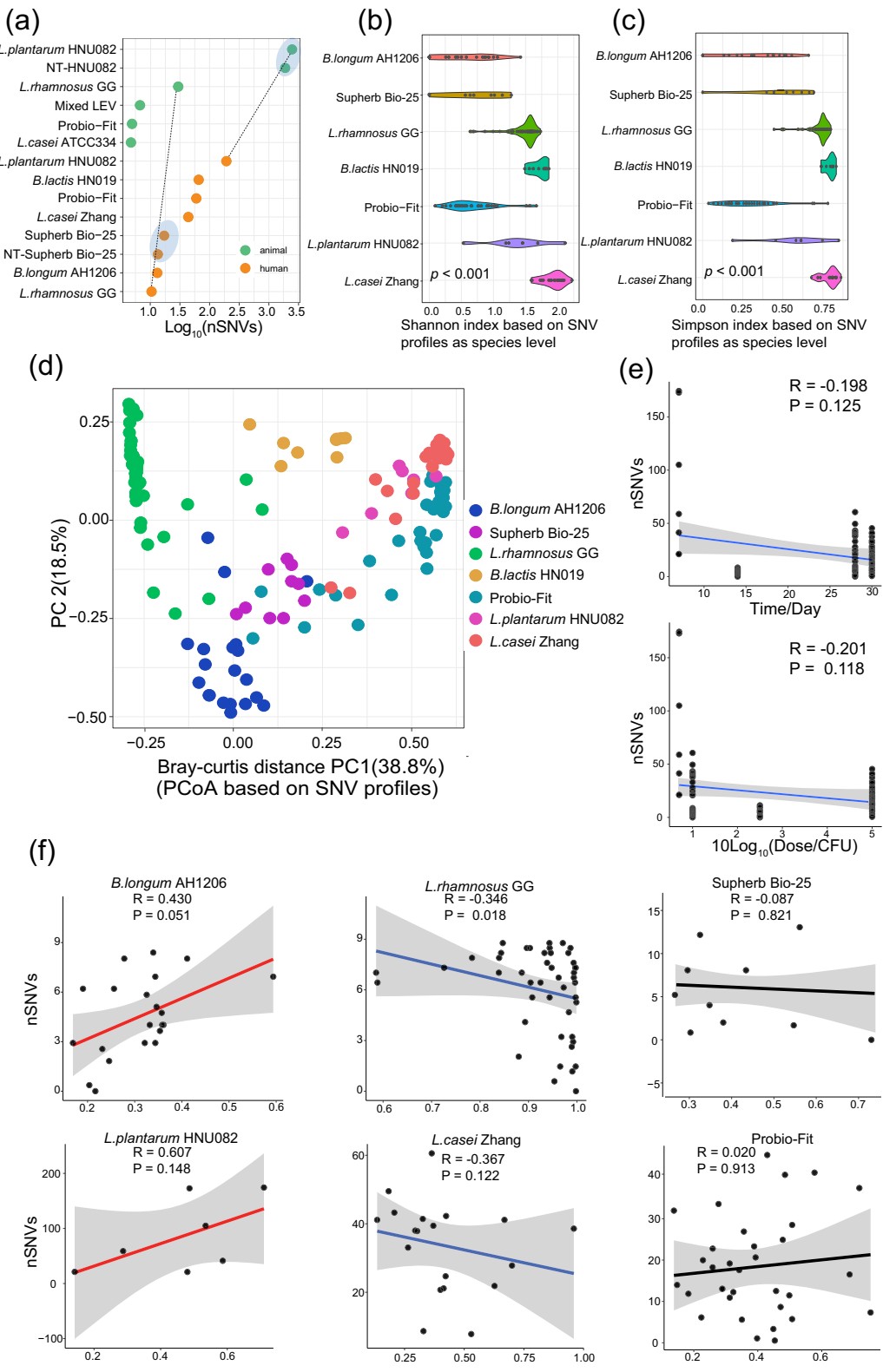

probiotic intervention (*B. longum* AH1206, $R = 0.43$; *L. plantarum* HNU082, $R = 0.607$), while *L. rhamnosus* GG and *L. casei* Zhang had a negative correlation (*L. rhamnosus* GG, $R = -0.346$; *L. casei* Zhang, $R = -0.367$). No correlation between the nSNVs caused by mixed probiotics and Bray−Curtis distance of gut microbiota was found. These suggested that the overall pattern of nSNVs induced by probiotics was highly probiotic-strain-specific.

**Universal adaptive mutations in indigenous gut microbes in response to probiotic intervention.** We identified three bacterial gut residents that accumulated the convergent genetic changes in response to probiotic consumption in six human metagenomic studies, including *Faecalibacterium prausnitzii* (*F. prausnitzii*), *Eubacterium rectale*, and *Roseburia intestinalis* (Supplementary Data 2 and Fig. 3a−c). Interestingly, the probiotic interventions

**Fig. 2 Evolutionary changes induced by probiotics and their associations with experimental factors. a** A scatter plot shows the normalized number of SNVs (i.e., SNV number/sequencing depth of the target genome) induced by probiotic intervention for all included studies. Dashed lines connect the same probiotic strains in the animal experiment and human cohorts. The blue shadow represents the comparison of nSNVs produced by probiotics with that of the control group (NT-HNU082 and NT-Superb Bio-25). **b, c** In human cohorts, alpha diversity indexes (Shannon and Simpson index) were calculated based on the species-level profile of SNVs induced by probiotic intake (P < 0.001). The shaded area indicates the full probability distribution of the variable. **d** In human cohorts, the genetic beta-diversity difference within and between studies was estimated based on the species-level SNV profile of native gut microbiomes. **e** No significant correlations between duration ($R = 0.134$, $P = 0.205$) and dose of probiotics ($R = -0.112$, $P = 0.289$) and the normalized number of SNVs induced by probiotics. **f** The study-dependent correlation between the Bray−Curtis distance of microbial species abundance profiles and the normalized number of SNVs induced by probiotics, including positive (AH1206 and HNU082), negative (LGG and Zhang), and uncorrelated (mixed probiotics). The source data for graphs are available as Supplementary Data 7 or FigShare (https://figshare.com/projects/Probiotic_consumption_ influences_universal_adaptive_mutations_in_indigenous_human_and_mouse_gut_microbiota/122447).

did not significantly alter the relative abundance of all these three species (Wilcoxon rank-sum test, $p > 0.05$), except for the LGG cohort (Wilcoxon rank-sum test, $p < 0.05$, *F. prausnitzii* and *Eubacterium rectale*). While ecological alterations in the gut microbiome were limited, the probiotic intervention led to widespread shifts in the genetic composition (detectable SNVs) of these individual gut residents (Fig. 4a, *F. prausnitzii*, $p = 0.026$). These suggested that evolutionary response might precede the ecological changes in the microbial communities under selection pressure.

To investigate if different probiotic interventions can lead to similar genomic variations, candidate adaptive SNVs were explored, which can be commonly found in at least three out of six probiotics-intervention studies. Remarkably, *F. prausnitzii* ATCC 27768 had the most shared SNVs ($N = 19$) across independent studies (Supplementary Data 4), while *Eubacterium rectale* and *Roseburia intestinalis* also had two shared SNVs respectively. We next validated whether these candidate adaptive SNVs produced by probiotic intervention can also occur in the control group (null model, Supplementary Data 5). The four SNVs from *Eubacterium rectale* and *Roseburia intestinalis* can be also identified in Israel control cohorts (null model). Two SNVs from *F. prausnitzii* in the probiotics group were detected in the control group as well. Therefore, we pinpointed a total of adaptive 17 SNVs occurred in *F. prausnitzii* specifically adapted to probiotic intake and can be validated across distinct host cohorts (Fig. 4a).

**Functional annotation of SNV-related genes of *F. prausnitzii* induced by probiotic intervention.** Among those 17 adaptive SNVs due to probiotics consumption, 13 (76.5%) occurred in the gene coding regions of functional genes. Seven were non-synonymous mutations, while six were synonymous mutations. These mutations involved in nine functional proteins, including 30S ribosomal protein S5, phosphohydrolase, sensor histidine kinase KdpD, ferritin, fprA family A-type flavoprotein, nitroreductase family protein, ribonucleotide-diphosphate reductase subunitbeta, peptidase S24 and Type II toxin-antitoxin system PemK/MazF family toxin (Fig. 4a and Table 3), including four types of mutations A > G ($n = 6$), A > C ($n = 6$), G > A ($n = 3$) and G > T ($n = 2$) (Fig. 4b and Table 3). Given six protein-expressing genes contained non-synonymous mutations. Next, Phyre2 was employed to predict the protein structure before and after probiotic intake and further visualized how these non-synonymous genetic mutations significantly changed the protein structure via EZMOL. The predicted structure of nitroreductase family protein and fprA family A-type flavoprotein has been substantially modified (Fig. 4c), suggesting significant changes in the functional potential of the gut microbiome after probiotic exposure. The structures and amino acid sequences of other proteins have been provided in Supplementary Fig. 2.

To investigate how differentially functional genes responded to the gut selective pressure due to probiotic intake, the ratio of non-synonymous and synonymous (dN/dS) was calculated. The dN/dS ratio < 0.25 indicated the purifying selection acting on the genes, while the ratio > 1 suggests that a gene was under positive selection for adapting to a new and or changing habitat[23,24]. In our study, the dN/dS ratios in different probiotic interventions ranged from 0.15 to 2.0 or from 0.25 to 1 (Fig. 4d). This suggested that different functional genes of a gut microbial strain can have diverse evolutionary trends. Moreover, the same gene may present parallel evolutionary trends under the different interventions of probiotics. Specifically, the dN/dS ratio of nitroreductase family protein was > 1 in probiotics *B. longum* AH1206, *L. plantarum* HNU082, and *L. casei* Zhang group. Phosphohydrolase was positively selected during the probiotic treatment with both *L. plantarum* HNU082 and mixed probiotics (Probio-Fit). Also, the same dN/dS ratios pattern for mixed probiotics (Probio-Fit) and a single-strain probiotic (*L. plantarum* HNU082) was exhibited in peptidase S24 and type II toxin-antitoxin system PemK/MazF family toxin. Nonetheless, different probiotic products may still have distinct patterns of evolutionary effect on a microbial functional gene of gut residents. Under the intervention of probiotic strain *L. plantarum* HNU082, the dN/dS ratios of ferritin and fprA family A-type flavoprotein were > 1, while the mixed strains intervention was the opposite. Notably, only one gene, sensor histidine kinase KdpD, was under purifying selection (dN/dS < 0.25). It suggests that most genes in *F. prausnitzii* tend to be neutral by the new gut environment shaped by the probiotic ingestion. The above results illustrated the distinct evolutionary changes in the intestinal microbiota under the environmental pressure of different probiotic interventions.

**The heritability of adaptive SNVs induced by probiotic intervention.** To investigate whether or how long such adaptive mutations accumulated in the key gut residents, such as *F. prausnitzii*, can be inherited, an independent longitudinal microbiome study of probiotic intervention was conducted using *L. plantarum* HNU082 as a model strain (Fig. 5a). All six human participants in this validation study successfully completed two experimental phases: (I) continuous probiotic intervention for 7 days; (II) a long-term follow-up microbiome study (6 months after phase I). They volunteered to provide stool samples throughout all experimental phases as requested. Firstly, we identified 610 SNVs of *F. prausnitzii* at the end point of phase I, while a total of 1828 SNVs genomes were identified at phase II. Among those 610 SNVs identified from phase I, 317 (51.96%) were transient mutations that were not detectable at phase II, while 293 (48.04%) were retained on the *F. prausnitzii* genome at phase II (Fig. 5b). These suggested that probiotic intervention led to long-lasting yet often overlooked genetic changes in the gut residents. In the 293 heritable SNVs and 317 transient SNVs we

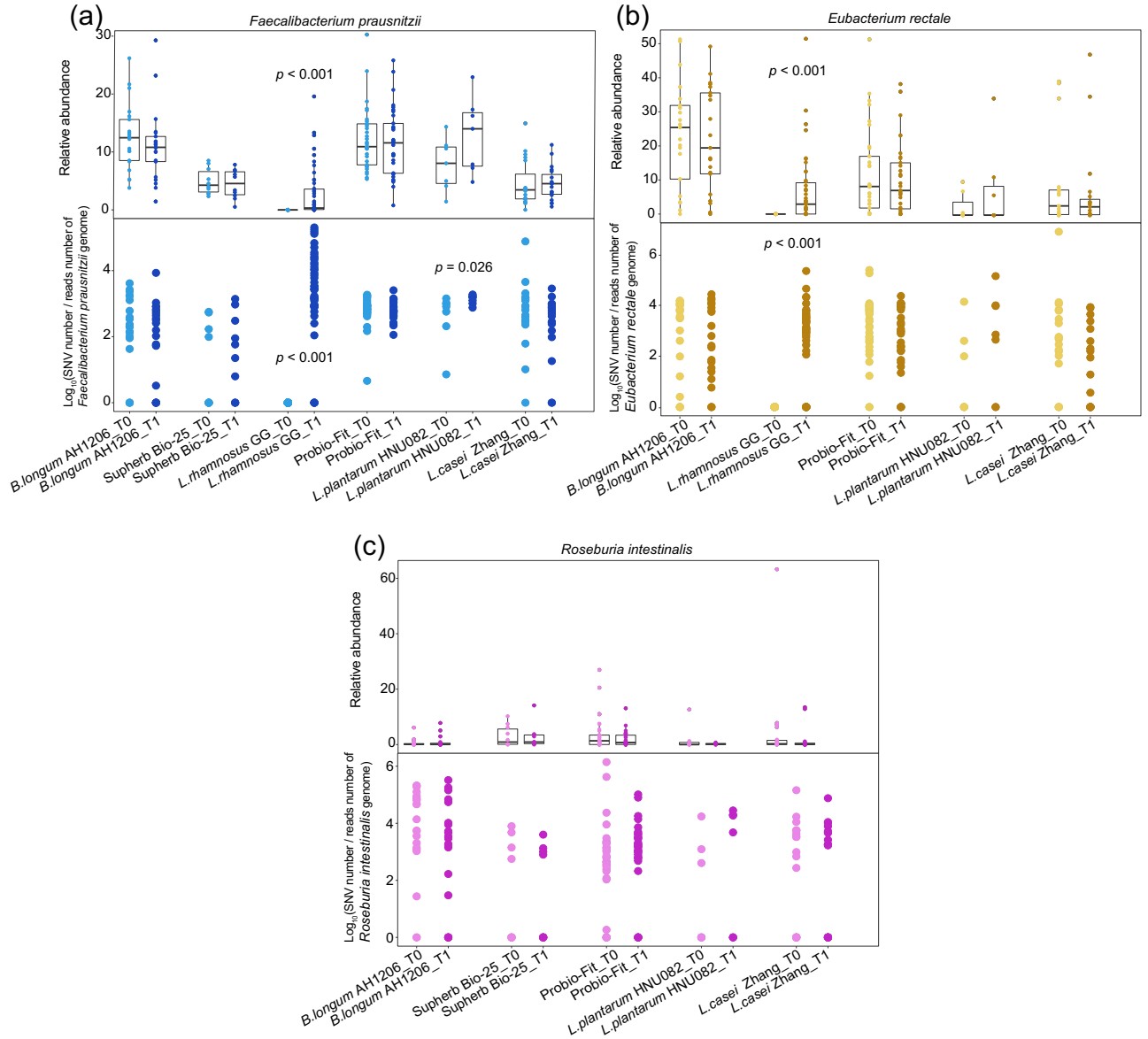

**Fig. 3 Linking ecological and evolutionary changes of three key gut resident species in response to probiotic intervention in multiple studies.** The boxplots indicate the relative abundance and the normalized number of SNVs of **a** *F. prausnitzii*, **b** *Eubacterium rectale*, **c** *Roseburia intestinalis* in between time points in each study. Wilcoxon rank-sum test was used to compare the abundance or nSNVs between time points and considered the significance at 0.05 levels. For each comparison, T0 represents the baseline phase and T1 represents the end point of the intervention phase (time variable). The box represents the 25–75th percentile, whiskers represent the full range, and the line represents the median value. The source data for graphs are available as Supplementary Data 7 or FigShare (https://figshare.com/projects/Probiotic_consumption_influences_universal_adaptive_mutations_in_indigenous_human_and_mouse_gut_microbiota/122447).

observed, 129 functional genes were identified. Within the 129 functional genes, 39 were uniquely from heritable SNVs, 43 were uniquely from transient SNVs, and 47 overlap (Fig. 5c and Supplementary Data 6).

We next characterized the functional genes with entirely inherited or transient SNVs induced by probiotic intervention from phase I to II. Sixteen entirely SNV-inherited proteins were identified firstly (Fig. 5d), which contained at least two consistent SNVs at both phases I and II. We next functionally annotated 20 protein products that have at least two transient SNVs at phase I whereas these two SNVs were not detectable at phase II (Fig. 5e and Supplementary Data 6). For example, one of those entirely SNV-inherited proteins, FprA family A-type flavoprotein, possesses ten SNVs induced by probiotic HNU082 that can inherit in an extraordinarily long period. Intriguingly, most transient-SNVs-

related proteins are involved in carbohydrate transport and metabolism, such as carbohydrate ABC transporter permease, carbohydrate ABC transporter substrate-binding protein and carbohydrate-binding protein. These suggested that residents in the gut microbial communities tended to adaptively evolve carbohydrate-related proteins for the short-term probiotic invasion.

## Discussion

It has been widely recognized that probiotics can modulate the composition and function of gut microbiota[25]. SNVs and structural variants of gut microbiota also have long been noted[26]. However, evolutionary changes in gut microbes due to probiotics intervention remain poorly characterized. The increasing attention had been brought to the high strain-specificity of probiotics[19,27] and host individuality in the probiotic efficacy[28],

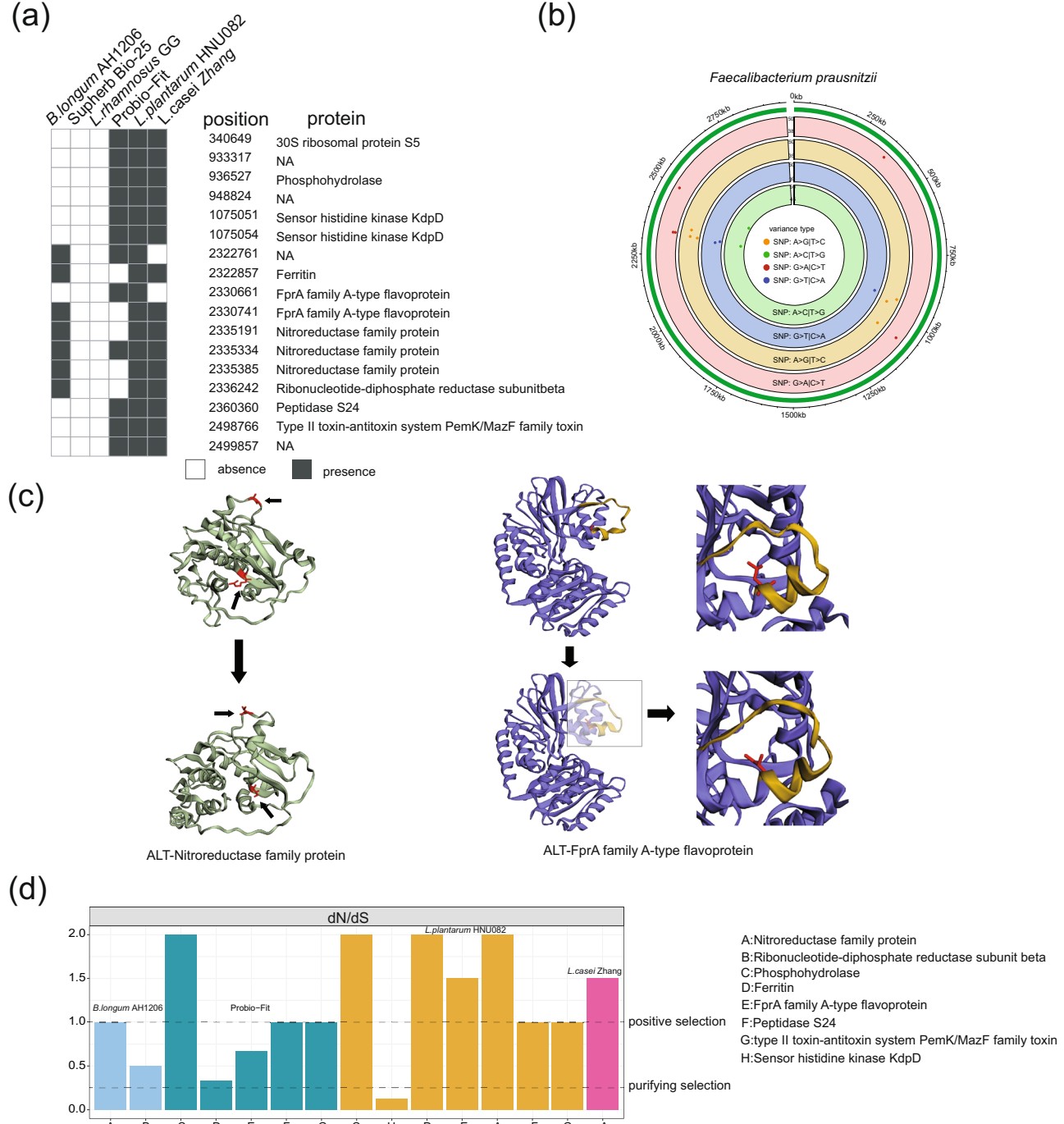

**Fig. 4 Functional annotation of SNVs occurred on *F. prausnitzii* ATCC 27768 induced by probiotic intervention and the calculation of dN/dS ratios.** **a** The details of 17 adaptive SNVs of *F. prausnitzii*. **b** The location of 17 unique SNVs in the genome of *F. prausnitzii*. All SNVs belonged to four types of mutations (A > G, A > C, G > A and G > T). **c** The crystal structure of Nitroreductase family protein and FprA family A-type flavoprotein. The red was the amino acid perssad. **d** The dN/dS ratios of genes with SNVs due to probiotics consumption (The dN/dS ratios can be performed when SNV has both non-synonymous and synonymous on the nine functional proteins.) The source data for graphs are available as Supplementary Data 7 or FigShare (https://figshare.com/projects/Probiotic_consumption_influences_universal_adaptive_mutations_in_indigenous_human_and_mouse_gut_microbiota/122447).

which motivated us to give priority to perform such a meta-analysis study. Hence, from the perspective of adaptive SNVs, our study assessed the effect of probiotic intake on the genomic stability of indigenous gut microbes, and we characterized the specific or common evolutionary changes of gut microbes under the selection pressure of a variety of probiotics.

The indigenous gut microbiome suffered increased intestinal selection pressure with the invasion of probiotics. Notably,

probiotics caused more adaptive mutations in gut microbiota than the control group, and more mutations were observed in mice than in humans. It suggested that there were strong antagonistic relationships between probiotics and indigenous gut microbes, which were more intense in mice. This is consistent with the results of the previous studies[22]. However, correlation analysis revealed that the number and magnitude of local adaptive SNVs were greatly related to the host environment, and

**Table 3 Summary of 19 SNVs of *F. prausnitzii*.**

| Position | Gene order | REF-base | ALT-base | REF-codon | ALT-codon | REF-aa | ALT-aa | Type | Protein |
|---|---|---|---|---|---|---|---|---|---|
| 340649 | 340 | C | T | GAG | AAG | E | K | N | 30 S ribosomal protein S5 |
| 933317 | NA | G | A | — | — | — | — | — | NA |
| 936527 | 909 | A | G | GAT | GAC | D | D | S | Phosphohydrolase |
| 948824 | NA | A | G | — | — | — | — | — | NA |
| 1075051 | 1048 | T | C | GAU | GAC | D | D | S | Sensor histidine kinase KdpD |
| 1075054 | 1048 | G | A | GAG | GAA | E | E | S | Sensor histidine kinase KdpD |
| 2322761 | NA | G | A | — | — | — | — | — | NA |
| 2322857 | 2245 | C | A | GAG | GAU | E | D | N | Ferritin |
| 2330661 | 2256 | A | G | GGU | GGC | G | G | S | FprA family A-type flavoprotein |
| 2330741 | 2256 | G | T | CUC | AUC | L | I | N | FprA family A-type flavoprotein |
| 2335191 | 2263 | G | A | CAC | CAU | H | H | S | Nitroreductase family protein |
| 2335334 | 2263 | C | T | GCC | ACC | A | T | N | Nitroreductase family protein |
| 2335385 | 2263 | G | A | CGC | UGC | R | C | N | Nitroreductase family protein |
| 2336242 | 2264 | T | C | GCA | GCG | A | A | S | Ribonucleotide-diphosphate reductase subunitbeta |
| 2360360 | 2288 | T | C | AAC | AGC | N | S | N | Peptidase S24 |
| 2498766 | 2293 | C | T | GCU | ACU | A | T | N | Type II toxin-antitoxin system PemK/MazF family toxin |
| 2499857 | NA | T | G | — | — | — | — | — | NA |
| 2359643 | NA | A | C | — | — | — | — | — | NA |
| 2360322 | 2288 | C | T | GAA | AAA | E | K | N | Peptidase S24 |

*N* non-synonymous, *S* synonymous.

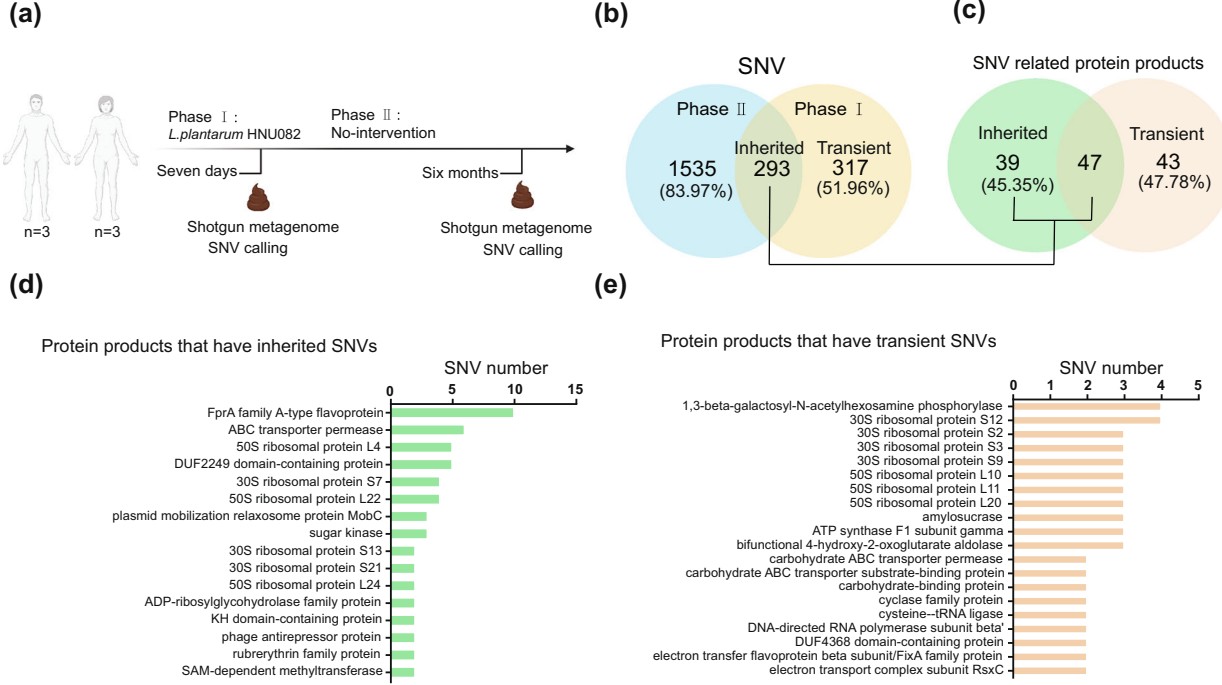

**Fig. 5 The heritability of SNVs from gut resident *F. prausnitzii* induced by probiotic intervention. a** Our experiment design of independent longitudinal microbiome study of probiotic intervention using Lp082 (*n* = 6). **b** Venn diagram indicates the unique and shared SNVs of *F. prausnitzii* induced by probiotic HNU082 in phase I (right after probiotic intervention) and II (6 months after probiotic intervention). **c** Venn diagram indicates the unique and shared proteins which inherited SNVs and transient SNVs involved. **d** The barplot indicates the SNV number of entirely SNV-inherited proteins. **e** The barplot indicates the SNV number of 20 protein products that had SNVs transiently occurred during the probiotic intervention. The source data for graphs are available as Supplementary Data 7 or FigShare (https://figshare.com/projects/Probiotic_consumption_influences_universal_adaptive_mutations_in_indigenous_human_and_mouse_gut_microbiota/122447).

which probiotic strain(s) have been supplemented. Accordingly, we hold the opinion that more studies with specific probiotic strains and various larger number populations should be needed to further explore the complicated relationships of probiotics and indigenous gut microbiota at the single-nucleotide level.

Identification of biomarkers is a key goal in clinical microbiome studies. Recently, evolutionary biomarkers (such as SNVs) of native gut microbes have been shown to be great indicators of host phenotypes[1,29]. Here, we presented the most comprehensive meta-analysis interrogating the universal evolutionary biomarkers

in the native gut microbiota that arose by probiotic treatment. Surprisingly, 17 adaptive SNVs commonly occurred on *F. prausnitzii* across multiple studies. *F. prausnitzii* is a well-known butyrate-producing bacterium as a potential probiotic for humans[30] by fermenting non-digestible carbohydrates[31]. The use of carbohydrates inevitably leads to competition between probiotics and gut microbes[18]. Encouragingly, universal adaptive mutations were observed in functional genes of *F. prausnitzii* genome involved in carbohydrate-related protein in the probiotic-intervention period, suggesting that *F. prausnitzii* was constantly adapting to the selection pressure of probiotics. Interestingly, nitroreductase was found to have positive evolution due to the consumption of probiotics *B. longum* AH1206, *L. plantarum* HNU082, and *L. casei* Zhang. Notably, probiotic strains modulated gut microbiota and microenvironment by enhancing fecal altered enzymes (nitroreductase), thus restoring histoarchitecture of the colon[32]. Therefore, the enhancement of nitroreductase may be the result of *F. prausnitzii* evolution under the selection pressure of probiotics. Further, a potentially beneficial mechanism of probiotics may be to decrease the nitroreductase activity of intestinal microbes under the selection pressure of probiotics to improve diseases, such as colorectal cancer[33,34]. Next, as a response, the dN/dS ratios of nitroreductase-related genes of gut microbes indicate the evolution of adapting to a new or changing habitat. Intriguingly, the sensor kinase KdpD showed a signal of purifying selection, which may not activate kdpFABC expression in the absence of KdpD. However, the kdpFABC can still be activated by cross-regulation (phosphohydrolase, the dN/dS ratio > 1)[35]. Overall, further work is needed to experimentally validate which adaptive variants can lead to direct loss or enhancement of protein function in either the short- or long-term run. Interestingly, as a result of the discontinuation of probiotics, hundreds of adaptive SNVs on genes primarily involved in carbohydrate-related proteins bounced back, suggesting dietary probiotic administration leads to a competition for nutrients with native gut microbes and a widespread but temporary adaptive evolution on the *F. prausnitzii* genome. Our results highlight genetic changes in *F. prausnitzii* under probiotics selective pressures that were not assumed before. In contrast, the other half of putative adaptive mutations can be observed for a long period (~6 months), which might lead to changes in bacterial functional capacity. These demonstrated that daily supplemental probiotics can form a powerful driver of the ecology and evolution of indigenous intestinal microbial communities, which has been often ignored[18]. Additionally, the personal probiotic history has never been considered for the assessment of the clinical outcome of the following probiotic treatments or other therapeutic treatments targeting on the gut microbiome. Therefore, our result provides insights into explaining the personalized efficacy of microbiome-targeted therapeutic treatments by consideration of the background genome structure or functional capacity of key gut residents based on probiotic-intake history.

Collectively, we found probiotics increased the instability of the gut microbial genome and highly divergent genomic responses to probiotics intake between humans and mice. Given the functional modules, the presence and absence of SNVs involving carbohydrate-related proteins suggest intensive competition between probiotics and gut microbes for carbon sources. This meta-analysis largely extended our understanding of the adaptive evolution of gut microbiota under the selection pressure of probiotics.

## Methods

### Sequence data collection and curation
A total of 1499 literature records were identified through the extensive database searching in PubMed and ISI Web of Science, while two records were kindly provided by peers. Next, 433 studies were retained after the removal of duplicates. The initial records were screened using keywords, titles, and abstracts, and 415 citations were excluded. Therefore, 18 studies were identified that we can get access to the full article and successfully performed shotgun metagenomic sequencing of stool samples collected from hosts that consumed probiotics. Among these 18 studies, seven studies were further filtered out as the corresponding sequencing data are not publicly accessible or its quality or sample size did not meet the minimum standard for re-analysis. Following the data curation process above (Supplementary Fig. 3), we finally pinpointed 11 probiotic studies, a total of 421 fecal samples, 224 probiotic-treatment individuals, and 197 placebo control that were included in our meta-analysis[7,8,10,18–20,22,36–38] (Tables 1 and 2). The study was approved by the Ethical Committee; for human participants, they provided informed consent before they enrolled in the study. Host models included mice, dogs, and rat and human cohorts spanning four countries (American, Israel, New Zealand, and China) with the administration of a single (*Lactobacillus* and *Bifidobacterium*) or mixed probiotic strains. All studies specifically aim to understand gut microbiome changes due to probiotics interventions, while no combined treatments related to prebiotics, drugs, etc. have been involved. In particular, *L. rhamnosus* GG and *L. plantarum* HNU082 were collected in both animal and human cohorts.

An unpublished cohort (probiotics *L. casei* Zhang), the sequence data have been deposited in the NCBI database (metagenomic sequencing data: PRJNA762428). In this study, we have recruited volunteers (ten females and ten males, BMI 18.98−21.54) who had an allergy history or not. The allergy was defined as: who suffered from had a severe allergic reaction due to one or more food and still allergic to it. The study was approved by the Ethical Committee of Hainan University, and informed consent was obtained from all volunteers before they enrolled in the study. They were asked to take probiotics tablets ($10^{10}$ CFU/day) for 28 days, and we collected their feces at baseline and at 28 days for metagenomic sequencing. Whole-genome shotgun sequencing of the samples was carried out using Illumina HiSeq 2500 instrument. Libraries were generated using a fragment length of approximately 300 bp. Paired-end reads were created using 150 bp in the forward and reverse directions.

### Quality control of the raw data and the removal of host DNA
Raw sra files were separated into paired or single fastq files using sratoolkit 2.10.7 software (https://github.com/ncbi/sra-tools). The raw reads were trimmed using Sickle (https://github.com/najoshi/sickle) and subsequently aligned to the host genome (human: GRCh38, mice: GRCm38.p6 dog: GCA_000002285.2, rat: GCA_000001895.4) to remove the host DNA fragments using Bowtie2 [39] with default settings.

### Identification of microbial taxonomy and SNV annotation
Firstly, MetaPhlan2 was employed to identify microbes and estimate their abundances in each stool sample using shotgun metagenomic sequencing reads[40]. The overall metagenomic sequencing depth and the sequencing coverage of each microbial strain can directly affect the identification of intestinal microbial SNVs. Therefore, based on the species-level profiles from MetaPhlan2, we pre-selected microbial species whose average relative abundance was greater than 0.5% for SNV annotation. The references or representative strains for all selected species from NCBI and their GenBank accessions are listed in Supplementary Data 1. Next, MIDAS[41] (Metagenomic Intra-Species Diversity Analysis System) was employed to profile the species-level SNV frequency and gene contents in the gut microbiota. Briefly, reference bacteria in a high-abundance genome database were constructed. Then, the shotgun metagenomic reads with 100 as minimum read depth were mapped to the database for SNV calling using Bowtie2 [39]. Candidate SNVs were identified and filtered with minimum quality 60 using SAMtools[42] and Bcftools (https://github.com/samtools/bcftools). For more details, refer to the code in the GitHub repository: https://github.com/HNUmcc/Probiotics-SNV-meta.

### Limited influence of different reference genomes on SNVs annotation
*F. prausnitzii* is the most common human gut microbe. In order to investigate the impact of reference genomes on our results, SNVs were annotated with multiple *F. prausnitzii* genomes with the methods we described previously. *F. prausnitzii* ATCC 27768 (NZ_CP030777, Assembly ID: GCF_003312465.1) was selected in our study as it is the top-1 representative reference genome recommended by NCBI. Next, additionally top-3 NCBI-recommended reference genomes were included for this species: *F. prausnitzii* A2165 (Assembly ID: GCF_002734145.1), *F. prausnitzii* JCM31915 (Assembly ID: GCF_010509575.1), *F. prausnitzii* Indica (Assembly ID: GCF_002586945.1). Firstly, we identified the presence of all four strains in the gut of BH1206 cohort and demonstrated the accumulated coverage (Supplementary Fig. 4). The coverage (%) of a reference genome on each sample was calculated and the relationship was visualized between the cumulative coverage and the number of metagenome samples included in a study. Both ×1 (blue) and ×100 (orange) minimum sequencing depth were considered for genome coverage calculation here. We found that the genome coverage of these genomes rapidly increased with multiple samples included, and the accumulated coverage almost saturated after less than ten metagenome samples were included. These suggested that all these included reference genomes can be detected and extensively covered by stool metagenome reads from most samples (Supplementary Data 4a).

Next, the genome-wide distance was compared between these four genomes with the average nucleotide identity (ANI) values (http://enve-omics.ce.gatech.edu/ani/index) (Supplementary Table 1). Typically, microorganisms that belong to the same species have over 95% ANI among themselves. However, the ANI values between NZ_CP03077 and other newly selected ones were far less than this conventional species boundary of ANI values (Supplementary Data 4b). Firstly, we tested if or how much percentage of these four genomes can be covered by the shotgun metagenomic reads from stool samples in a human cohort (e.g., BH1206).

Again, MIDAS was employed to profile the species-level SNV frequency and gene contents. We next compared the SNVs annotation results with different reference genomes on the four cohorts, including BH1206, Bio-25, LGG, Probio-Fit, HNU082 and Zhang (Supplementary Figs. 5, 6). Firstly, with our SNVs calling pipeline, no significant difference was found in the number of nSNVs between the different *F. prausnitzii* reference genomes at the T0 and T1 time points in the vast majority of studies and only slight differences were found in the BH1206 and Probio-Fit cohorts (Supplementary Data 4c). Secondly, the gene functions affected by SNVs changes between T0 and T1 time points (or due to probiotic intervention) were largely similar (Supplementary Data 4d). This indicated that it is plausible and sufficient to select ATCC 27768 as the reference genome.

**Definition of adaptive SNV induced by probiotic intervention.** In this study, only the SNV profiles were investigated in the native gut microbiome, while insertions and deletions were not our focus. The mutant quality of a base (produced by Bcftools) below 60 is excluded. In this manuscript, the paired data were focused (baseline and end point of probiotic consumptions) in the population cohort. Unmatched animal studies and human cohorts ($N = 6$) were not included in the meta-analysis (starting at Fig. 2b). Next, as illustrated in Supplementary Data 5, adaptive mutations that occurred after probiotic consumption do not necessarily relate to nucleotides on the reference genome. We mapped the metagenome reads from the same hosts at time points to the same reference genomes and identified single-nucleotide changes (adaptive SNVs) before and after the probiotic consumption. Next, candidate adaptive SNVs due to probiotic consumptions were thought to meet the following requirements. (1) For a given microbial species (genome), a single-nucleotide difference should be identified between baseline and end point of a host, despite the nucleotide difference between the reference genome and either of them (Supplementary Fig. 7). (2) Such a genetic change can be observed in at least 30/50% of hosts in a study (Supplementary Fig. 8). (3) Such a genetic change did not show up within a period that is not related to any probiotic treatments for a host. Ideally, we can further exclude SNVs that are not adaptive, when they met the requirement (1) but also showed up before the time points of a host consumed the probiotic. However, most studies did not sample the time points before probiotic treatment except for the Israeli cohort (Bio-25). Therefore, for this study, we specifically removed such SNVs as they are less likely to be related to probiotic consumption. These excluded SNVs are mainly located at *Megamonas rupellensis*, *Roseburia inulinivorans*, *Roseburia intestinalis*, *Eubacterium rectale*, etc. After the correction by null model, a reasonable set of adaptive SNVs results was obtained in Supplementary Data 2. We further finalized the set of universal adaptive SNVs which can be detected in at least three (50%) of six studies with the technical requirements as we mentioned before.

**Statistics and reproducibility.** All statistical analyses were performed using R software. The differential abundances of various profiles were tested with the Wilcoxon rank-sum test, and the significant difference was considered at a nominal level of $p < 0.05$. Alpha diversity analysis was performed by in-house R code. Beta-diversity analysis was conducted using "vegan" and "plyr" package, and PCoA based on Bray−Curtis dissimilarity matrix was used to visualize the sample clustering on gut microbial composition. The package "ggplot" was used to generate boxplot, barplot, violin plot, and fitted curve. The heatmap was constructed using the "pheatmap" package. The packages, "circlize", "ComplexHeatmap", and "grid" were used for SNV genome circle map. The protein structure was predicted and displayed using Phyre2[43] (http://www.sbg.bio.ic.ac.uk/phyre2/html/page.cgi?id=index) and EZMOL[44] (http://www.sbg.bio.ic.ac.uk/ezmol/). The Venn diagram is rendered by using InteractiVenn[45] (http://www.interactivenn.net/).

## Data availability
The sequencing data could be downloaded from NCBI (Tables 1 and 2 show the accession numbers), and all data are available from the corresponding authors. The source data for graphs are available as Supplementary Data 7, and all source data, supplementary materials and SNVs density plots for B. longum AH1206 cohort have been deposited at FigShare (https://figshare.com/projects/Probiotic_consumption_influences_universal_adaptive_mutations_in_indigenous_human_and_mouse_gut_microbiota/122447). Any other information can be obtained from the corresponding authors upon reasonable request.

## Code availability
All code for data analysis in this project are available at the Github repository: https://github.com/HNUmcc/Probiotics-SNV-meta. https://github.com/Deeeeen/microbiome_SNV_calling.

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

## Acknowledgements

This work was supported by the Key Research and Development Project of Hainan Province (no. 320RC513) and the National Natural Science Foundation of China Program (no. 31871773).

## Author contributions

The study was designed by J.Z., Q.Z. and S.H. Data collection and analysis were performed by C.M., C.Z., D.C., S.J., S.S. and D.H. The manuscript was written by C.M., J.Z., S.H., D.C. and Q.Z. All authors read and approved the final manuscript.

## Competing interests

The authors declare no competing interests.
