## [Transparent Peer Review File · Communications Biology]

Reviewers' comments:

Reviewer #1 (Remarks to the Author):

This manuscript addresses the question of if consumption of probiotics can lead to evolutionary changes in the resident microbiota. This is an important question and answering it is not an easy task. The authors use a meta-analysis to determine if probiotic consumption is associated with variations of strains in the gut. The authors also propose to identify adaptive mutations linked to probiotics.

The paper is not very well written and the limitations of the approach taken to tackle the questions posed are not discussed. The most critical issues to me are that:

- the authors do not explain how they decide if a SNP is adaptive or not as no rejection of what is expected under a neutral model is considered (see e.g. analysis of evolution in human microbiomes in Garud et al 2019 PLoS Biology, Zhao et al 2019 Cell Host Microbe)
- it is not clear how the results shown depend on the reference genome used.
- at many instances of the manuscript it is not clear if the authors think that probiotics are mutagenic mutations or more reasonably cause a change in selection pressures which could alter the polymorphism patterns of the resident bacteria species.

The methods of the paper were also just briefly described, and do not make clear the statistical analysis done in the metagenomic data, and how they allow to support that certain SNPs are adaptive. For example on page 16 section definition of SNV induced by probiotic intervention it is not clear for how many individuals there were samples before and after the treatment and what the authors assume to be induced mutations.

Thus, I felt that the methods and analysis presented are not sufficient to support many of the inferences written in the paper.

Reviewer #2 (Remarks to the Author):

Reviewer comments

The manuscript by Ma et al., describes the nucleotide mutations in the gut microbiota that can occur due to probiotic supplementation. Although a relatively novel concept and interesting study idea the current manuscript has a number of areas which are not clearly described and need much improvement.

1. Revision of the text required. There are a number of grammatical errors, colloquial language in places and the use of "we" in too many instances. (Replace with It was found. It was investigated where possible).
2. The Introduction needs more specific examples from the literature of coevolution, genomic variation and selective evolution pressure. Even if not with probiotics but other pressures and how bacterial SNVs evolve.
3. Methods: Just 11 final studies is a small number given the number initially investigated, it is not completely clear why these were chosen. Also, the inclusion of the animal cohorts needs clarification and justification given the difference in their microbiomes and intestinal tract?
4. Results: DN/DS ratios needs a lot more attention in the results section, this is very important aspect to the study in terms of evolutionary pressures.

The specific genetic loci that are under pressure has not been developed in the manuscript.

There is a trial mentioned in the results section which includes 6 people. Is this a specific probiotic trial included in this study? The details of this is not described in the materials and methods (trial/supplement/length of treatment/ethical approval). I acknowledge there is brief referral to it in the supplemental methods.

5. Discussion does not fully evaluate the SNVs and the potential consequences of them. Again, the potential significance of the DN:DS ration results needs expanding on here. The impact on F.

prausnitzii is interesting and well explained.
6. Figure 1 could be supplementary data.

COMMSBIO-21-0733-T

Universal adaptive mutations of indigenous gut microbiota due to probiotics consumption

Dear editor and reviewers:

We really appreciate that you gave the chance for us to improve our manuscript. We have carefully revised our manuscript based on your professional and insightful comments. You may kindly find the point-to-point responses to the comments in the following text.

Reviewer #1

This manuscript addresses the question of if consumption of probiotics can lead to evolutionary changes in the resident microbiota. This is an important question and answering it is not an easy task. The authors use a meta-analysis to determine if probiotic consumption is associated with variations of strains in the gut. The authors also propose to identify adaptive mutations linked to probiotics.

Response: Thank you for your interest on our study and for raising some concerns. We apologize for the unclear words and misleading statements in the Methods section. Please find our point-to-point responses below.

The paper is not very well written and the limitations of the approach taken to tackle the questions posed are not discussed. The most critical issues to me are that: -the authors do not explain how they decide if a SNP is adaptive or not as no rejection of what is expected under a neutral model is considered (see e.g. analysis of evolution in human microbiomes in Garud et al 2019 PLoS Biology, Zhao et al 2019 Cell Host Microbe)

Response: We appreciate this reviewer brought up this critical point as well as a few important references in this field. Agree, we might detect “adaptive evolution” in bacteria by the independent recurrence of similar mutations in genes under natural selection or by an increase in mutational frequency that is inconsistent with neutral drift [1]. Therefore, it is critical to add null model for identifying the adaptive SNPs in the probiotic experiments. As suggested, we added the reference in Introduction section (line 80-82, page 4). Garud et al focused on the gut species that were quasi-phaseable at consecutive time points in longitudinally sampled HMP subjects. They attempted to resolve a single nucleotide difference between two of these time points in a genome-wide scan, while considering between-host difference. Similarly, in this manuscript, we also analyzed the six host-paired metagenomic datasets (from baseline to end time point of probiotic consumptions). Host-unmatched studies (N=6) were not included (starting from Figure 2B). To address this issue, we firstly resolve the single nucleotide difference between two time points for each host. Next, we understand SNVs can

randomly occurred between hosts. As considering SNVs under the neutral model, we need to exclude the SNVs that are significantly different between hosts. Last but not least, in this meta-analysis, combined the six longitudinal metagenome studies to explore universal adaptive mutations that can attributed to probiotic consumption. Collectively, if a SNV is “adaptive”, it should meet following technical requirements (1) for a given microbial species (genome), a single nucleotide difference should be identified between baseline and end point of a host, despite the nucleotide difference between the reference genome and either of them; (2) such a genetic change can be observed in at least 30% of human hosts in a study; (3) such a genetic change can be also found in three (50%) of six studies.

Adaptive parallel evolution also can be confirmed by Figure 4D. The figure 4D indicated different genes of the same gut microbes have different evolutionary trends, and the same gene may present parallel evolutionary strategies under the intervention of different probiotics. Specifically, the dN/dS ratio of nitroreductase family protein was no less than 1 in probiotics *B.longum* AH1206, *L.plantarum* HNU082, and *L.casei* Zhang group. Positive selection of phosphohydrolase were found in both *L.plantarum* HNU082 and mixed probiotics (Probio-Fit). Also, the same dN/dS ratios pattern for mixed probiotics (Probio-Fit) and single probiotic (*L.plantarum* HNU082) was shown in Peptidase S24 and Type II toxin-antitoxin system PemK/MazF family toxin.

In this meta-analysis, we focused on *Faecalibacterium prausnitzii* and demonstrated that SNVs of *Faecalibacterium prausnitzii* were adaptive parallel evolution. So, if we can, we want to keep "adaptive evolution" in the title.

In this revised manuscript, we further considered the null model. However, the ideal null model is that fecal samples were also collected before the baseline, and only the cohort in Israel met this condition. Therefore, we subtracted SNVs from the data for Israel in its neutral model. We found that SNVs in the neutral model belongs to *Megamonas rupellensis*, *Roseburia inulinivorans*, *Roseburia intestinalis*, *Eubacterium rectale*, etc. We have modified the number of SNVs in Supplemental material 2 and Figure 1-2.

Indeed, line 180-193, page 9: “To investigate if different probiotics interventions can induce the consensus genomic variations, we explored candidate adaptive SNVs, which can be commonly found in at least three out of six probiotics-intervention studies. Remarkably, *Faecalibacterium prausnitzii* ATCC 27768 had the most shared SNVs (N=19) across independent studies (Supplemental material 3, Table 2), while *Eubacterium rectale* and *Roseburia intestinalis* also had two shared SNVs respectively (Supplemental material 3). We next validated whether these candidate adaptive SNVs produced by probiotic intervention can also occur in the control group (null model). The four SNVs from *Eubacterium rectale* and *Roseburia intestinalis* can be also identified in Israel control cohorts (null model). Two SNVs from *Faecalibacterium prausnitzii* in probiotics group were detected in control group as well. Therefore, we pinpointed a total of adaptive 17 SNVs occurred in *Faecalibacterium prausnitzii* specifically adapted to probiotic intake and can be validated across distinct host cohorts (Figure 4A, Supplemental material 4).” The results indicated that an increase

inmutational frequency on *Faecalibacterium prausnitzii* genome that is inconsistent with neutral drift and 17 SNVs occurred in *Faecalibacterium prausnitzii* was adaptive.

1. Shijie Zhao, Tami D. Lieberman, Mathilde Poyet, Kathryn M. Kauffman, Sean M. Gibbons, Mathieu Groussin, Ramnik J. Xavier, Eric J. Alm. Adaptive Evolution within Gut Microbiomes of Healthy People[J]. Cell Host & Microbe, 2019, 25(5).

-it is not clear how the results shown depend on the reference genome used.

Response: Thank you for raising this point. Initially, we also considered possible effects due to the selection of reference genomes. However, we hope the following points can dispel your concerns.

1. In theory, the selection of representative reference genomes does not affect the identification of adaptive SNVs in longitudinally sampled hosts.

Firstly, in this paper, we mainly focus on the host-paired metagenome data (measured at the baseline and end time point of probiotic consumption) in the population cohort. Other studies that only have unmatched animal or human subjects were not included in the meta-analysis (starting from Figure 2B). Next, as illustrated in the figure below, “adaptive” mutations occurred after probiotic consumption do not necessarily related to what nucleotides on the reference genome. We mapped the metagenome reads from the same hosts at timepoints to the same reference genomes, and identified single-nucleotide changes on the genome before and after the probiotic consumption. Assuming a nucleotide on the reference genome is “A”, the possible change from the baseline to end point can be “A-to-T”, “A-to-G”, “A-to-C” due to probiotic consumption. Inversely, at the baseline, this nucleotide was divergent (“T”, “G” or “C”) from that on the reference genome (“A”) while it was mutated backward to “A” after probiotic consumption, which possibly resulted in three mutation types: “T-to-A”, “G-to-A”, and “C-to-A”. Collectively, the adaptive SNV would rather depend on the nucleotide changes between time points. Therefore, we reason that observed genetic changes (i.e., SNVs) wouldn’t affect by the selection of reference genomes.

2. Representative but distinct reference genomes of *Faecalibacterium prausnitzii* were commonly used in previous studies.

The selection of representative reference genomes is indeed critical to the SNV results, however, hasn't been adequately considered in the many papers. To our knowledge, *Faecalibacterium prausnitzii* is a relatively complex taxonomic groups that includes a huge number of genomes (N>80), while six are complete genomes available from NCBI. Furthermore, the reference genome database is still expanding. Several tools (such as MIDAS[1], and metaSNV [2]) commonly used for identifying SNPs from shotgun metagenome data, often include 1-3 representative reference genomes. We believe it is hard to make the judgement on which reference genome should be included in such studies.

Other hallmark works in this field also employed the single representative genome for certain species to identify SNVs, such as *Zhao et al 2019 Cell Host Microbe* etc. [3-7]. We basically followed the similar mapping strategy as they did.

1. Nayfach, S., Rodriguez-Mueller, B., Garud, N. & Pollard, K. S. An integrated metagenomics pipeline for strain profiling reveals novel patterns of bacterial transmission and biogeography. *Genome Res.* 26, 1612–1625 (2016).

2. Costea, P. I. et al. metaSNV: A tool for metagenomic strain level analysis. *PLOS ONE* 12, e0182392 (2017).

3. Shijie Zhao, Tami D. Lieberman, Mathilde Poyet, Kathryn M. Kauffman, Sean M. Gibbons, Mathieu Groussin, Ramnik J. Xavier, Eric J. Alm. Adaptive Evolution within Gut Microbiomes of Healthy People[J]. *Cell Host & Microbe*, 2019, 25(5).

4. Idan Yelin, Kelly B. Flett, Christina Merakou, Preeti Mehrotra, Jason Stam, Erik Snesrud, Mary Hinkle, Emil Lesho, Patrick McGann, Alexander J. McAdam, Thomas J. Sandora, Roy Kishony, Gregory P. Priebe. Genomic and epidemiological evidence of bacterial transmission from probiotic capsule to blood in ICU patients[J]. *Nature Medicine*, 2019, 25.

5. Chen Yaowen, Li Zongcheng, Hu Shuofeng, Zhang Jian, Wu Jiaqi, Shao Ningsheng, Bo Xiaochen, Ni Ming, Ying Xiaomin. Gut metagenomes of type 2 diabetic patients have characteristic single-nucleotide polymorphism distribution in *Bacteroides coprocola*. [J]. *Microbiome*, 2017, 5(1).

6. Zhu Qiyun, Hou Qiangchuan, Huang Shi, Ou Qianying, Huo Dongxue, VázquezBaeza Yoshiki, Cen Chaoping, Cantu Victor, Estaki Mehrbod, Chang Haibo, BeldaFerre Pedro, Kim HoCheol, Chen Kaining, Knight Rob, Zhang Jiachao. Compositional and genetic alterations in Graves' disease gut microbiome reveal specific diagnostic biomarkers. [J]. *The ISME journal*, 2021.

7. Ma Chenchen, Wasti Sanjeev, Huang Shi, Zhang Zeng, Mishra Rajeev, Jiang Shuaiming, You Zhengkai, Wu Yixuan, Chang Haibo, Wang Yuanyuan, Huo Dongxue, Li Congfa, Sun Zhihong, Sun Zheng, Zhang Jiachao. The gut microbiome stability is altered by probiotic ingestion and improved by the continuous supplementation of galactooligosaccharide. [J]. *Gut microbes*, 2020, 12(1).

3. Benchmark results on multiple *Faecalibacterium prausnitzii* genomes

3.1 Four representative *Faecalibacterium prausnitzii* genomes were selected

In order to investigate the impact of reference genomes on our results, we annotated SNVs using multiple *Faecalibacterium prausnitzii* genomes with the methods as we described previously. *Faecalibacterium prausnitzii* ATCC27768 (NZ_CP030777,

Assembly ID: GCF_003312465.1) was selected in our study as it is the top-1 representative reference genome recommended by NCBI. Next, we additionally included three that were top-3 NCBI-recommended reference genomes for this species: *Faecalibacterium prausnitzii* A2165 (Assembly ID: GCF_002734145.1), *Faecalibacterium prausnitzii* JCM31915 (Assembly ID: GCF_010509575.1), *Faecalibacterium prausnitzii* Indica (Assembly ID: GCF_002586945.1).

Next, we compared genome-wide distance between these four genomes with the Average Nucleotide Identity (ANI) values (<http://enve-omics.ce.gatech.edu/ani/index>). Typically, micro-organisms that belong to the same species have over 95% ANI among themselves. However, the ANI values between NZ_CP03077 and other newly selected ones were far less than this conventional species boundary of ANI values. Nonetheless, we felt that these *Faecalibacterium prausnitzii* genomes can be representative within species and helpful to address the reviewer's comment.

ANI value	NZ_CP03077 7	NZ_CP02247 9	NZ_CP02381 9	NZ_CP04843 7
NZ_CP03077 7	100%			
NZ_CP02247 9	86.24%	100%		
NZ_CP02381 9	86.73%	97.39%	100%	
NZ_CP04843 7	86.38%	99.99%	97.41%	100%

Firstly, we tested if or how much percentage these four genomes can be covered by the shotgun metagenomic reads from stool samples in a human cohort (e.g., BH1206). We calculated the coverage (%) of a reference genome on each sample and visualized the relationship between the accumulative coverage and the number of metagenome samples included in a study. Both 1X (blue) and 100X (orange) minimum sequencing depth for were considered for genome coverage calculation here. We found that the genome coverage of these genomes rapidly increased with multiple samples included, and the accumulated coverage almost saturated after less than 10 metagenome samples included. These suggested that all these included reference genomes can be detected and extensively covered by stool metagenome reads from most samples. **(Supplemental material 11A or figure below).**

3.2 The SNV results with ATCC 27768 was highly reproducible with other selected reference genomes

We next compared the SNVs annotation results with different reference genomes on the four cohorts, including BH1206, Bio-25, LGG, Probio-Fit, HNU082 and Zhang.

Firstly, with our SNVs calling pipeline, we found no significant difference in the number of nSNVs between the different *F. prausnitzii* reference genomes at the T0 and T1 time points in the vast majority of studies, and only slight differences were found in the BH1206 and Probio-Fit cohorts (**Supplemental material 11C or boxplots below**). SNVs density plots for BH1206 were provided for more details (**Supplemental material 12**). Secondly, the gene functions affected by SNVs changes between T0 and T1 time points (or due to probiotic intervention) were largely similar (**Supplemental material 11D or venn diagrams below**). This indicated that it is plausible and sufficient to select ATCC27768 as the reference genome.

B1_gene SNVs Gene Products

B2_gene SNVs Gene Products

B3_gene SNVs Gene Products

B4_gene SNVs Gene Products

B5_gene SNVs Gene Products

B6_gene SNVs Gene Products

(B1: BH1206, B2: Bio-25, B3: LGG, B4: Probio-Fit, B5: HNU082, B6: Zhang)

We have formally added the above analysis results to supplemental materials and inserted the details into the Methods section. Overall, we concluded that the selection of genomes showed very limited effects on the SNV annotated results.

Collectively, the evolutionary study of resident gut microbes is novel yet challenging, especially in the field of probiotics. We have been trying to avoid the as many technical biases as possible in order to reach more reliable biological conclusions. According to our test with a panel of reference genomes from the same species *Faecalibacterium prausnitzii*, we believe that SNV results reported (*Faecalibacterium prausnitzii* ATCC 27768) would not significantly change as the different selection of reference genomes. Regarding the reviewer's concern, we specifically highlighted what reference genome was used from identifying SNV changes induced by probiotic ingestion in the revised manuscript.

-at many instances of the manuscript it is not clear if the authors think that probiotics are mutagenic mutations or more reasonably cause a change in selection pressures which could alter the polymorphism patterns of the resident bacteria species.

Response: Thanks for the reviewer proposing these very intriguing hypotheses. To our knowledge, it is still challenging to distinguish whether the genetic polymorphism of the indigenous gut microbe is caused by probiotics directly interacting with the indigenous gut microbes or by probiotics indirectly altering the gut selection pressure of the host. We here focused on the metagenomic datasets on probiotics which is not able to answer this question. But it is indeed a very interesting topic, and we need to design a new study to explore this mechanism in the near future. The meta-analysis will be a good start. Overall, we believe that the gut passage of probiotic can trigger a series of complex interactions that can change the genetics of both probiotic itself and a wide array of indigenous gut microbiota. From this paper, most conclusions are more than related to associations.

Notably, we recently published a paper on Microbiome journal [1], which might in part explain this question. In this study, we demonstrated that *Lactobacillus plantarum* (Lp082) can apply a highly convergent adaptation strategy under diverse host selection pressures (i.e., humans, mice and zebrafish) yet differentially influence the resident gut microbiota. Interestingly, resident gut microbial strains, especially competing strains with Lp082 (e.g., *Bacteroides* spp. and *Bifidobacterium* spp.), actively responded to Lp082 engraftment by accumulating 10–70 times more evolutionary changes than usual.

To avoid any potential confusion, "probiotic-induced" in the manuscript were corrected as "due to probiotics consumption".

[1]Huang Shi,Jiang Shuaiming,Huo Dongxue,Allaband Celeste,Estaki Mehrbod,Cantu Victor,BeldaFerre Pedro,VázquezBaeza Yoshiki,Zhu Qiyun,Ma Chenchen,Li Congfa,Zarrinpar Amir,Liu YangYu,Knight Rob,Zhang Jiachao. Candidate probiotic *Lactiplantibacillus plantarum* HNU082 rapidly and convergently evolves within human, mice, and zebrafish gut but differentially influences the resident microbiome.[J]. *Microbiome*,2021,9(1).

-The methods of the paper were also just briefly described, and do not make clear the statistical analysis done in the metagenomic data, and how they allow to support that certain SNPs are adaptive. For example on page 16 section definition of SNV induced by probiotic intervention it is not clear for how many individuals there were samples before and after the treatment and what the authors assume to be induced mutations.

Response: We appreciate your insightful suggestions. We apologize for the unclear statements in the Methods section. Please find our point-to-point responses below.

1. The statistical support of adaptive SNV caused by probiotic intervention

In this revised manuscript, we carefully clarify how we identified adaptive SNVs due to probiotic consumption. Explicitly, if a SNV was “adaptive”, it should meet following technical requirements.

(1) For a given microbial species (genome), a single nucleotide difference should be identified between baseline and end point of a host, despite the nucleotide difference between the reference genome and either of them. (2) Such a genetic change can be observed in at least 30-50% of hosts in a study. (3) Such a genetic change didn't show up within a period that are not related to any probiotic treatments for a host. Ideally, we can further exclude SNVs that are not “adaptive”, when they met the requirement (1) but also showed up before the time points of a host consumed the probiotic. However, most studies didn't sample the time points before probiotic treatment except for the Israeli cohort (Bio-25). Therefore, for this study, we specifically removed such SNVs that are not fully related to probiotic consumption. We found that SNVs in the neutral model mainly located at *Megamonas rupellensis*, *Roseburia inulinivorans*, *Roseburia intestinalis*, *Eubacterium rectale*, etc. We have modified the number of SNVs in Supplemental material 2 and Figure 1-2.

We finalized the set of 17 universal adaptive SNVs which can be detected in at least three (50%) of six studies with the technical requirements as we mentioned before.

As suggested, we further updated results accordingly, line 180-193, page 9:

“To investigate if different probiotics interventions can induce the consensus genomic variations, we explored candidate adaptive SNVs, which can be commonly found in at least three out of six probiotics-intervention studies. Remarkably, *Faecalibacterium prausnitzii* ATCC 27768 had the most shared SNVs (N=19) across independent studies (Supplemental material 3, Table 2), while *Eubacterium rectale* and *Roseburia intestinalis* also had two shared SNVs respectively (Supplemental material 3). We next validated whether these candidate adaptive SNVs produced by probiotic intervention can also occur in the control group (null model). The four SNVs from *Eubacterium rectale* and *Roseburia intestinalis* can be also identified in Israel control cohorts (null model). Two SNVs from *Faecalibacterium prausnitzii* in probiotics group were detected in control group as well. Therefore, we pinpointed a total of adaptive 17 SNVs occurred in *Faecalibacterium prausnitzii* specifically adapted to probiotic intake and can be validated across distinct host cohorts (Figure 4A, Supplemental material 4).”

The results indicated that the increase in mutational frequency on *Faecalibacterium prausnitzii* genome that is inconsistent with neutral drift and 17 SNVs occurred in *Faecalibacterium prausnitzii* was adaptive.

2. Number of host subjects in the six human cohorts used to identify adaptive SNVs

To mitigate the potential effect of confounding factors (such as individuality in gut microbiome) on our analysis, we next focused on six human-related studies that have host-paired metagenome data before and after the probiotic intervention (*B.longum* AH1206, Supherb Bio-25, *L.rhamnosus* GG, Probio-Fit, *L.plantarum* HNU082 and *L.casei* Zhang).

The raw datasets were too large and have been transferred to <https://github.com/HNUMcc/Probiotics-SNV-meta/pull/1/files>. We showed the SNV profiles of all paired individuals in six cohorts (BH1206: 21 individuals, Bio-25:10 individuals, LGG: 42 individuals, Probio-Fit: 33 individuals, HNU082:7 individuals, Zhang: 19 individuals). The supplementary material contains the number of individuals, SNVs information, etc. The custom scripts used for the SNV analysis can be found on GitHub:

https://github.com/Deeeeen/microbiome_SNV_calling

3. Correction of the word "induced"

We're sorry that the use of "probiotic-induced" was confusing, which has been corrected to "due to probiotics consumption". Reviewer is correct. We agree that, at present, we are still unable to according to the current meta-analysis.

-Thus, I felt that the methods and analysis presented are not sufficient to support many of the inferences written in the paper.

Response: Thank you for raising these concerns about methods. We apologize for any unclear and misleading statements in the Methods section. As mentioned earlier, we have substantially modified our methods and analysis. Please feel free to let us know if any improvement needed.

Reviewer #2

Reviewer comments

The manuscript by Ma et al., describes the nucleotide mutations in the gut microbiota that can occur due to probiotic supplementation. Although a relatively novel concept and interesting study idea the current manuscript has a number of areas which are not clearly described and need much improvement.

Response: Thank you for your interest and positive comments on our study and giving us insightful suggestions. We have sent our manuscript to the AJE Company for language editing service to improve the readability. Also, we carefully improved the details of the methods to increase the readability and reproducibility of the results. In addition, we extended the references to the introduction and discussion sections. You can kindly find the point-to-point responses in the following text.

1. Revision of the text required. There are a number of grammatical errors, colloquial language in places and the use of “we” in too many instances. (Replace with It was found. It was investigated where possible).

Response: Thank you for raising this point. We've modified them as suggested and checked/revised the potential grammar errors in our manuscript carefully.

2. The Introduction needs more specific examples from the literature of coevolution, genomic variation and selective evolution pressure. Even if not with probiotics but other pressures and how bacterial SNVs evolve.

Response: We appreciate your insightful suggestions. Thanks to our continuous interest in the co-evolution of probiotics and indigenous gut microbiota, we can add a few specific examples here from our recent published papers. Overall, probiotics are live micro-organisms and typically leverage different strategies to adapt within the gut under various selective pressures, e.g., spontaneous adaptive mutations. Our recent study (<https://doi.org/10.1186/s40168-021-01102-0>) clearly showed how a candidate probiotic *Lactiplantibacillus plantarum* HNU082 rapidly and convergently evolves within human, mice, and zebrafish gut, while differentially influences the resident microbiome [1]. Furthermore, in another recent study from our lab, we demonstrated that continuous prebiotics supplement with probiotics can significantly affect the SNV patterns of both probiotics and indigenous gut microbiota of mice [2]. Other than these, we also added references reported by other labs on the such a topic, such as *E coli*. *Nissle*.

You can find the updated Introduction paragraph and references.

We further updated the Introduction and now the revised part reads:

Previous study has highlighted that the evolution and transfer of genetic information of host-associated microbiota could enable resilience to biotic and abiotic perturbations [2]. Take *Bacteroides fragilis* as an example, many parallel evolution of genes were found related in cell-envelope biosynthesis and polysaccharide utilization [3]. Notably, probiotic can impose persistent selective pressures on host gut bacteria by accumulating mutations related to carbohydrate utilization and acid tolerance within the mouse gut

microbiome, such as *E. coli*. Nissle [4], and the driving force may provide the potential for genomic variations of gut resident species [5]. The indigenous gut microbiota, including competitors and collaborators, rapidly evolved to adapt the ecological invasion of probiotics [6]. However, these *in vivo* genetic processes of gut microbiota are still poorly characterized due to probiotic consumption using a wide array of human and animal models. The *in vivo* evolution of the indigenous gut microbiota and probiotics facilitates the understanding to leverage these gut selective forces for genetic engineering of probiotics. (Page 3-4, line 64-77)

Adaptive evolution of gut microbes can be confirmed by parallel evolution or convergent evolution or by increased frequency of mutations inconsistent with neutral drift [5]. (Page 4, line 80-82)

1. Huang Shi, Jiang Shuaiming, Huo Dongxue, Allaband Celeste, Estaki Mehrbod, Cantu Victor, Belda Ferre Pedro, Vázquez Baeza Yoshiki, Zhu Qiyun, Ma Chenchen, Li Congfa, Zarrinpar Amir, Liu Yang Yu, Knight Rob, Zhang Jiachao. Candidate probiotic *Lactiplantibacillus plantarum* HNU082 rapidly and convergently evolves within human, mice, and zebrafish gut but differentially influences the resident microbiome. [J]. *Microbiome*, 2021, 9(1).

2. Ferreira A, Crook N, Gasparrini AJ, Dantas G. Multiscale Evolutionary Dynamics of Host-Associated Microbiomes. *Cell* 2018; 172:1216-27.

3. Zhao S, Lieberman TD, Poyet M, Kauffman KM, Gibbons SM, Groussin M, et al. Adaptive Evolution within Gut Microbiomes of Healthy People. *Cell Host Microbe* 2019; 25:656-67 e8.

4. Crook N, Ferreira A, Gasparrini AJ, Pesesky MW, Gibson MK, Wang B, et al. Adaptive Strategies of the Candidate Probiotic *E. coli* Nissle in the Mammalian Gut. *Cell Host Microbe* 2019; 25:499-512 e8.

5. Zhao S, Lieberman TD, Poyet M, Kauffman KM, Gibbons SM, Groussin M, et al. Adaptive evolution within gut microbiomes of healthy people. 2019; 25:656-67. e8.

6. Ma C, Wasti S, Huang S, Zhang Z, Mishra R, Jiang S, et al. The gut microbiome stability is altered by probiotic ingestion and improved by the continuous supplementation of galactooligosaccharide. *Gut Microbes* 2020; 12:1785252.

3. Methods: Just 11 final studies is a small number given the number initially investigated, it is not completely clear why these were chosen. Also, the inclusion of the animal cohorts needs clarification and justification given the difference in their microbiomes and intestinal tract?

Response: We apologize that the original manuscript fell short in clarifying how we screened and included studies in the meta-analysis in the Methods section.

Our inclusion and exclusion criteria for the 11 selected studies include:

1. The study has a longitudinal design, which at least has a baseline and end time point for the probiotic consumption for a human or animal host subject.

2. The study doesn't use probiotics in combination with any other substance, such as medications, prebiotics, minerals, vitamins.

3. The study's raw data were published and had detailed metadata.

4. The study's sequencing data quality allows us to analyze at least species-level composition in the gut microbiome.

5. The study provided clear probiotics species/strains/product information.
6. The study has a clear statement on the dose and duration for probiotic intake.

For the animal cohorts, inclusion and exclusion criteria were consistent with the population cohorts. We agree that significant biological differences can be found among animal microbiomes and intestinal tract. However, we still felt interested in whether adaptive SNVs in the indigenous gut microbiome of animals can be observed at the level/degree as we did on humans. Next, we expected to compare the SNV frequency (or even diversity) caused by the same probiotics among different host species (e.g., mice and human) as we did in another study (<https://doi.org/10.1186/s40168-021-01102-0>). Due to the small number of animal trials, in this paper, we do not have direct evidence to explain the relationship of SNVs number among different host species. As many animal studies didn't track the metagenome of hosts longitudinally, our subsequent analysis excluded them. We demonstrated that a variety of probiotic consumption can lead to widespread single nucleotide variants (SNVs) in the native microbiota of animal gut.

4. Results: DN/DS ratios needs a lot more attention in the results section, this is very important aspect to the study in terms of evolutionary pressures.

The specific genetic loci that are under pressure has not been developed in the manuscript.

There is a trial mentioned in the results section which includes 6 people. Is this a specific probiotic trial included in this study? The details of this is not described in the materials and methods (trial/supplement/length of treatment/ethical approval). I acknowledge there is brief referral to it in the supplemental methods.

Response: We appreciate the reviewer brought up this critical point. We've performed additional analysis of dN/dS of all related genetic mutations and updated the Results sections accordingly.

“The dN/dS ratio < 0.25 indicated a purifying selection acting on the genes, while the ratio > 1 suggests that a gene was under positive selection for adapting to a new and or changing habitat”. (Line 216-218, page 10)

“This suggested that different functional genes of a gut microbial strain can have diverse evolutionary trends. Moreover, the same gene may present parallel evolutionary trends under the different interventions of probiotics. Specifically, the dN/dS ratio of nitroreductase family protein was > 1 in probiotics *B.longum* AH1206, *L.plantarum* HNU082, and *L.casei* Zhang group. Phosphohydrolase were positively selected during the probiotic treatment with both *L.plantarum* HNU082 and mixed probiotics (Probio-Fit). Also, the same dN/dS ratios pattern for mixed probiotics (Probio-Fit) and a single-strain probiotic (*L.plantarum* HNU082) was exhibited in Peptidase S24 and Type II toxin-antitoxin system PemK/MazF family toxin. Nonetheless, different probiotic products may still have distinct patterns of evolutionary effect on a microbial functional gene of gut residents.” (Line 220-230, page 11)

“Notably, only one gene, sensor histidine kinase KdpD, was under purifying selection (dN/dS < 0.25). It suggests that most genes in *Faecalibacterium prausnitzii* tend to be positively selected by the new gut environment shaped by the probiotics

ingestion.” (Line 233-236, page 11)

Figure 5A was designed to illustrate this independent dataset/study (Human-probiotics *Lactobacillus plantarum* HNU082) including six human subjects. We'd like to use this study to externally validate if these adaptive SNVs identified in the meta-analysis can be heritable. This study has been now published and mainly described the universal adaptive evolution strategy of probiotic *Lactobacillus plantarum* (Lp082) to gut selection pressure from humans, mice and zebrafish [1]. Supplementary material 7 has included the details on this study. Supplementary material 7 also includes another study (Human-probiotics *Lactobacillus casei* Zhang) that has not yet been published. The aforementioned two studies were "additional records identified through other sources (such as peer labs)" (Figure S10). You can find more details about two trials in supplementary material 7.

[1]Huang Shi,Jiang Shuaiming,Huo Dongxue,Allaband Celeste,Estaki Mehrbod,Cantu Victor,BeldaFerre Pedro,VázquezBaeza Yoshiki,Zhu Qiyun,Ma Chenchen,Li Congfa,Zarrinpar Amir,Liu YangYu,Knight Rob,Zhang Jiachao. Candidate probiotic *Lactiplantibacillus plantarum* HNU082 rapidly and convergently evolves within human, mice, and zebrafish gut but differentially influences the resident microbiome.[J]. *Microbiome*,2021,9(1).

Supplementary materials 7:

(Human-probiotics *Lactobacillus plantarum* HNU082):

The experimental design

*For human participants, each individual was informed of the experimental guidelines and details and consent obtained; seven volunteers (four females and three males, BMI 19.19-22.49) agreed to participate in the experiment. During the experiment, the subjects were asked to avoid ingesting any probiotic product or antibiotic and to maintain their regular diet. They did not have inflammatory bowel disease or diabetes and had not used antibiotics for at least 3 months prior to sampling. They were asked to consume vacuum freeze-drying *Lactobacillus plantarum* HNU082 powder (including 7×10^9 CFU live strains) 2g every day for 7 days. Six months after they stopped taking the probiotics, six volunteers (three females and three males) provided stool samples again. So, seven healthy participants finished the probiotics consumptions experiment. Interesting, six healthy participants finished the whole experiment, which were studied on the heritability of adaptive SNVs induced by probiotic intervention.*

Ethical approval

The study was reviewed and approved by the Ethics Committee of Hainan University (HNU-2018037, Haikou, China), and informed consent was obtained from all volunteers before they enrolled in the study. The participants provided written informed consent to participate in the study. Sampling and all described subsequent steps were conducted in accordance with the approved guidelines.

Metagenomic DNA extraction and shotgun metagenomic sequencing and data quality control

The QIAamp® DNA Stool Mini Kit (Qiagen, Hilden, Germany) was used for DNA

extraction from the fecal samples. The quality of the extracted DNA was assessed by 0.8% agarose gel electrophoresis, and the OD 260/280 was measured by spectrophotometry. All of the DNA samples were subjected to shotgun metagenomic sequencing by using an Illumina HiSeq 2500 instrument in the Novogene Company (Beijing, China). Libraries were prepared with the paired-end reads (2 x 150 bp). The raw reads were trimmed using Sickle (<https://github.com/najoshi/sickle>) and subsequently aligned to the human genome to remove the host DNA fragments.

(Human-probiotics *Lactobacillus casei* Zhang)

The experimental design

In this study, we have recruited volunteers (ten females and ten males, BMI 18.98-21.54) who had the allergy history or not. The allergy was defined as: who suffered from had a severe allergic reaction due to one or more food and still allergic to it. They were asked to take probiotics tablets (10^{10} CFU/day) for 28 days, and we collected their feces at baseline and at 28 days for metagenomic sequencing.

Shotgun metagenomic sequencing and quality control

DNA extraction was carried out using CWBIO Stool Genomic DNA Kit (CW2092, CWBIO, China) as per the manufacturer's guidelines. Whole-genome shotgun sequencing of the samples was carried out using Illumina HiSeq 2500 instrument. Libraries were generated using a fragment length of approximately 300 bp. Paired-end reads were created using 150 bp in the forward and reverse directions. The reads were trimmed using Sickle and were subsequently aligned to the human genome with the reference genome (hg38 database) to remove the host DNA fragments.

Ethical approval

The study was reviewed and approved by the Ethics Committee of Zhejiang Gongshang University, and informed consent was obtained from all volunteers before they enrolled in the study. The participants provided written informed consent to participate in the study. Sampling and all described subsequent steps were conducted in accordance with the approved guidelines.

5. Discussion does not fully evaluate the SNVs and the potential consequences of them. Again, the potential significance of the DN:DS ration results needs expanding on here. The impact on *F. prausnitzii* is interesting and well explained.

Response: We thank reviewer making this insightful suggestion. We've now added discussion on the SNVs and the dN/dS ratio, in line 310-323, page 15, which reads:

Interestingly, nitroreductase was found to have positive evolution due to the consumptions of probiotics *B.longum* AH1206, *L.plantarum* HNU082 and *L.casei* Zhang. Notably, probiotic strains modulated gut microbiota and microenvironment by enhancing fecal altered fecal enzymes (nitroreductase), thus restored histoarchitecture of the colon [1]. Therefore, the enhancement of nitroreductase may be the result of *Faecalibacterium prausnitzii* evolution under the selection pressure of probiotics. Further, a potentially beneficial mechanism of probiotics may be to decrease the nitroreductase activity of intestinal microbes under selection pressure of probiotics to improve diseases, such as colorectal cancer [2, 3]. Next, as a response, the dN/ds ratios

of nitroreductase-related genes of gut microbes indicate the evolution of adapting to a new or changing habitat. Intriguingly, the gene (Sensor histidine kinase KdpD) was purifying selection (the dN/dS ratio less than 0.25), which may not activate kdpFABC expression in the absence of KdpD. However, the kdpFABC can still be activated by cross-regulation (Phosphohydrolase, the dN/dS ratio greater than 1) [4].

1. Chandel D, Sharma M, Chawla V, Sachdeva N, Shukla G. Isolation, characterization and identification of antigenotoxic and anticancerous indigenous probiotics and their prophylactic potential in experimental colon carcinogenesis. *Sci Rep* 2019; 9:14769.
2. de Moreno de LeBlanc A, Perdigon G. Reduction of beta-glucuronidase and nitroreductase activity by yoghurt in a murine colon cancer model. *Biocell* 2005; 29:15-24.
3. Verma A, Shukla G. Probiotics *Lactobacillus rhamnosus* GG, *Lactobacillus acidophilus* suppresses DMH-induced procarcinogenic fecal enzymes and preneoplastic aberrant crypt foci in early colon carcinogenesis in Sprague Dawley rats. *Nutr Cancer* 2013; 65:84-91.
4. Schramke H, Laermann V, Tegetmeyer HE, Brachmann A, Jung K, Altendorf K. Revisiting regulation of potassium homeostasis in *Escherichia coli*: the connection to phosphate limitation. *Microbiologyopen* 2017; 6.

6. Figure 1 could be supplementary data.

Response: It is a great suggestion. We've moved the Figure 1 to Figure S10. This supplementary figure showed the flow chart of literature screening and data curation process.

The scripts involved in the revision can be found in github:
https://github.com/Deeeeen/microbiome_SNV_calling
<https://github.com/HNUMcc/Probiotics-SNV-meta>

Again, thank you and the anonymous reviewers for the constructive comments that have improved our manuscript. If you have any additional suggestions, please do not hesitate to contact me via email.

Yours sincerely,
Jiachao Zhang
zhjch321123@163.com

College of Food Science and Engineering, Key Laboratory of Food Nutrition and Functional Food of Hainan Province, Hainan University, Haikou, 570228, China

REVIEWERS' COMMENTS:

Reviewer #1 (Remarks to the Author):

The authors have made substantial changes in the manuscript that addressed reasonably well my previous points.

I have some further suggestions for improvement as follows:

pg3 line 55: consider writing: evolutionary pressures leading to changes, such as ...

pg 3 line 64 consider writing: Previous studies have highlighted...

pg 6 lines 118 and 122 and pg 7 line 133: please add the statistical test done to address the significance.

pg7 line 149: typo does should be dose.

pg 9 line 180 considering changing "can induce the consensus genomic variations" to can lead to similar genomic variations.

pg 10 line 205-206. Please re-write the sentence because it is not clear to understand the meaning.

pg 11 line 235 tend to be neutral, instead of tend to be positively selected.

pg 12 lines 254-255 please consider removing the part of the sentence "which can profoundly impact the taxonomic compositions and metabolic functions of the gut microbiome" to avoid speculative statements.

pg14 line 286 "increasing instability of the gut microbiome" is too excessive.

pg 15 line 313 fecal altered enzymes; instead of fecal altered fecal enzymes.

pg 15 line 320 remove parenthesis and better to write: the sensor kinase KdpD showed a signal of purifying selection (...).

pg 16 line 330 Consider writing : Our results highlight genetic changes in *Faecalibacterium prausnitzii* under probiotics selective pressures that were not assumed before. In contrast, the other half of putative adaptive mutations can be observed for a long period (~6 months), which might lead to changes in bacterial functional capacity.

pg 20 lines 421-423 Please re-write to remove and help to address the reviewer.

pg 20 line 426 cumulative, not accumulative.

pg 21 lines 452-454. Please re-write as the sentence is difficult to understand.

pg 22 line 468 please consider writing: we specifically removed such SNVs as they are less likely to be related to probiotic consumption.

Reviewer #2 (Remarks to the Author):

Thank you for addressing the comments raised in the first review. A few minor points below:

Overall:

- "we" is still used too much throughout the text.

- Some colloquial terms still in use which need revising.

- *Faecalibacterium prausnitzii* after initial use can then be written as *F. prausnitzii*

Additionally:

Line 50: Change probiotic to probiotics

Line 59: Remove "the" (before host health)

Line 64: Insert "A" before previous study

Line 66: re word to "In the organism *Bacteroides fragilis*, for example, ..."

Line 68: Change probiotic to probiotics (and throughout where the plural is required)

Line 99: Change metagenome to metagenomic

Line 101: Change doesn't to does not

Line 151-152: References need to be included

Line 157 and elsewhere: Space required between bacterial genus and species

Line 305 and 328: remove the word "fierce"
Line 337: Change "Especially" to "Additionally"
Line 340: Change "provided" to "provides"

COMMSBIO-21-0733-A

Universal adaptive mutations of indigenous gut microbiota due to probiotics consumption

Dear reviewers:

We really appreciate that you gave the chance for us to improve our manuscript. We have carefully revised our manuscript based on your professional and insightful comments. You may kindly find the point-to-point responses to the comments in the following text.

Reviewer #1

The authors have made substantial changes in the manuscript that addressed reasonably well my previous points. I have some further suggestions for improvement as follows:

Response: Thank you for your interest and positive comments on our study and giving us insightful suggestions. We have carefully revised our manuscript according to your comments. You can kindly find the point-to-point responses to your comments in the following text.

pg3 line 55: consider writing: evolutionary pressures leading to changes, such as ...

Response: We appreciate your helpful comment. We have modified the sentence (page 3, line 50).

pg 3 line 64 consider writing: Previous studies have highlighted...

Response: We apologize for this oversight. We have modified it (page 3, line 60).

pg 6 lines 118 and 122 and pg 7 line 133: please add the statistical test done to address the significance.

Response: We appreciate your insightful comment. We have added the statistical test in page 6, line 121-122 and page 6, line 131 as “Interestingly, the number of gut resident species occurring SNVs significantly decreased with hosts after the dietary intervention with Probio-Fit, *L.rhamnosus* GG and *B.lactis* HN019, besides in the mice with *L.plantarum* HNU082 (Wilcoxon rank-sum test, Figure 1a) and The consumption of probiotic *L.plantarum* HNU082, *L.rhamnosus* LGG and *B.lactis* HN019 significantly reduced the total frequency of SNVs (nSNVs) in the gut residents (Wilcoxon rank-sum test, Figure 1c).”

pg7 line 149: typo does should be dose.

Response: We apologize for this oversight. We have modified it (page 7, line 146).

pg 9 line 180 considering changing "can induce the consensus genomic variations" to can lead to similar genomic variations.

Response: We appreciate your helpful comment. We have modified the sentence (page 9, line 178-179).

pg 10 line 205-206. Please re-write the sentence because is it not clear to understand the meaning.

Response: We apologize for this oversight. We have edited the sentence as “Given six protein-expressing genes contained non-synonymous mutations” (page 10, line 202-203). Non-synonymous mutations can lead to changes in protein structure. So, we employed Phyre2 to predict the protein structure before and after probiotic intake and further visualized how these non-synonymous genetic mutations significantly changed the protein structure via EZMOL. The predicted structure of above protein can be found in Figure 4c and Supplementary figure 2.

pg 11 line 235 tend to be neutral, instead of tend to be positively selected.

Response: We appreciate your helpful comment. We have modified it (page 11, line 231).

pg 12 lines 254-255 please consider removing the part of the sentence "which can profoundly impact the taxonomic compositions and metabolic functions of the gut microbiome" to avoid speculative statements.

Response: We appreciate your helpful comment. We have removed it (page 12, line 248-249).

pg14 line 286 "increasing instability of the gut microbiome" is to excessive.

Response: We appreciate your helpful comment. We have modified the sentence as “The indigenous gut microbiome suffered increased intestinal selection pressure with the invasion of probiotics” (page 13, line 279-280).

pg 15 line 313 fecal altered enzymes; instead of fecal altered fecal enzymes.

Response: We apologize for this oversight. We have modified it (page 14, line 306).

pg 15 line 320 remove parenthesis and better to write: the sensor kinase KdpD showed a signal of purifying selection (...).

Response: We appreciate your helpful comment. We have modified the sentence as “Intriguingly, the sensor kinase KdpD showed a signal of purifying selection” (page 15, line 313-314).

pg 16 line 330 Consider writing : Our results highlight genetic changes in Faecalibacterium prausnitzii under probiotics selective pressures that were not assumed before. In contrast, the other half of putative adaptive mutations can be observed for a long period (~6 month), which might lead to changes in bacterial functional capacity.

Response: We appreciate your insightful suggestions. We have modified the sentence as “Our results highlight genetic changes in *F. prausnitzii* under probiotics selective pressures that were not assumed before. In contrast, the other half of putative adaptive mutations can be observed for a long period (~6 month), which might lead to changes in bacterial functional capacity” (page 15, line 322-326).

pg 20 lines 421-423 Please re-write to remove and help to address the reviewer.

Response: We apologize for this oversight. We have removed the sentence.

pg 20 line 426 cumulative, not accumulative.

Response: We apologize for this oversight. We have modified it (page 19, line 412).

pg 21 lines 452-454. Please re-write as teh seance is difficult to understand.

Response: We appreciate your insightful suggestions. We have re-write the sentence as “Next, as illustrated in Supplementary data 5, adaptive mutations occurred after probiotic consumption do not necessarily related to what nucleotides on the reference genome” (page 21, line 445-447).

pg 22 line 468 please consider writing: we specifically removed such SNVs as they are less likely to be related to probiotic consumption.

Response: We appreciate your insightful suggestions. We have modified the sentence as “Therefore, for this study, we specifically removed such SNVs as they are less likely to be related to probiotic consumption” (page21, line 460-461).

Reviewer #2

Thank you for addressing the comments raised in the first review. A few minor points below:

Response: Thank you for your interest and positive comments on our study and giving us insightful suggestions. We have carefully revised our manuscript according to your comments. You can kindly find the point-to-point responses to your comments in the following text.

Overall:

-“we” is still used too much throughout the text.

Response: Thank you for raising this point. Some sentences have been modified (page 2, line 27-38, 32-33; page 6, line 110, 124-125; page 7, line 139-141, 148; page 9, line 179; page 10, line 203; page 12, line 250, 255-256; page 14, line 295-296, 300; page 16, line 348-349; page 18, line 383, 391,393; page 19, line 402, 406, 411; page 20, 420, 428,431; page 21, line 440, 442-443; page 22, line 464).

-Some colloquial terms still in use which need revising.

Response: We appreciate your insightful suggestions. We have modified some sentence (page line).

- *Faecalibacterium prausnitzii* after initial use can then be written as *F. prausnitzii*

Response: We appreciate your insightful suggestions. We have used abbreviation as “*F. prausnitzii*”.

Additionally:

Line 50: Change probiotic to probiotics

Response: We apologize for this oversight. We have modified it (page 3, line 46).

Line 59: Remove “the” (before host health)

Response: We appreciate your insightful suggestions. We have modified it (page 3, line 54).

Line 64: Insert “A” before previous study

Response: We appreciate your insightful comment. We have modified it (page 3, line 60).

Line 66: re word to “In the organism *Bacteroides fragilis*, for example,

Response: We appreciate your insightful suggestions. We have modified it (page 3, line 62).

Line 68: Change probiotic to probiotics (and throughout where the plural is required)

Response: We apologize for this oversight. We have modified it (page 3, line 64).

Line 99: Change metagenome to metagenomic

Response: We apologize for this oversight. We have modified it (page 5, line 95).

Line 101: Change doesn't to does not

Response: We apologize for this oversight. We have modified it (page 5, line 97).

Line 151-152: References need to be included

Response: We appreciate your insightful suggestions. We have added references.

Line 157 and elsewhere: Space required between bacterial genus and species

Response: We apologize for this oversight. We double checked and corrected it in full manuscript.

Line 305 and 328: remove the word "fierce"

Response: We appreciate your insightful suggestions. We have removed it.

Line 337: Change "Especially" to "Additionally"

Response: We appreciate your insightful comment. We have modified it (page 15, line 328).

Line 340: Change "provided" to "provides"

Response: We appreciate your insightful comment. We have modified it (page 16, line 331).

Again, thank the anonymous reviewers for the constructive comments that have improved our manuscript. If you have any additional suggestions, please do not hesitate to contact me via email.

Yours sincerely,

Jiachao Zhang

zhjch321123@163.com

College of Food Science and Engineering, Key Laboratory of Food Nutrition and Functional Food of Hainan Province, Hainan University, Haikou, 570228, China